# Differences in visually induced MEG oscillations reflect differences in deep cortical layer activity

Dimitris A. Pinotsis [1,2] ✉ & Earl K. Miller[2]

Neural activity is organized at multiple scales, ranging from the cellular to the whole brain level. Connecting neural dynamics at different scales is important for understanding brain pathology. Neurological diseases and disorders arise from interactions between factors that are expressed in multiple scales. Here, we suggest a new way to link microscopic and macroscopic dynamics through combinations of computational models. This exploits results from statistical decision theory and Bayesian inference. To validate our approach, we used two independent MEG datasets. In both, we found that variability in visually induced oscillations recorded from different people in simple visual perception tasks resulted from differences in the level of inhibition specific to deep cortical layers. This suggests differences in feedback to sensory areas and each subject's hypotheses about sensations due to differences in their prior experience. Our approach provides a new link between non-invasive brain imaging data, laminar dynamics and top-down control.

[1] Centre for Mathematical Neuroscience and Psychology and Department of Psychology, City —University of London, London EC1V 0HB, UK. [2] The Picower Institute for Learning & Memory and Department of Brain and Cognitive Sciences, Massachusetts Institute of Technology, Cambridge, MA 02139, USA. ✉email: pinotsis@mit.edu

A major challenge in treating neurological diseases and disorders is the their heterogeneity[1]. The same symptom can arise from multiple causes and the same cause can lead to various symptoms. Only a small fraction of patients might respond to a certain drug, which leads to frustration for both patients and clinicians. One way to address this challenge is to build computational models of brain dysfunction[2]. This approach has given rise to a new field, computational psychiatry[3]. Instead of focusing on symptomatology, computational models focus on describing brain neurobiology and its alterations in patient brains. However, computational models are not a *panacea*: they are limited by the spatial and temporal scale of the dynamics they describe. Spiking and compartmental models describe single neurons and firing rates while neural mass models describe large brain networks and population activities. Thus, they are limited because neurological diseases and disorders involve interactions of many factors that are *simultaneously* expressed at multiple scales. They depend on both genetic variations and environmental factors spanning microscopic and macroscopic scales ranging from, for example, altered mitochondria and single neuron function to neuroinflammation of axons connecting different brain areas[4]. Here, we present a new way to link multiple scales. We use combinations of computational models to describe both macro- and microscales.

We combine a model of neural compartments, describing dendrites and somata introduced in ref. [5], with a biophysical neural mass model that predicts non-invasive brain data[6]. The model of ref. [5] had been used to explain magnetoencephalography (MEG) oscillations in the alpha/beta[7] and gamma bands[8]. We used statistical decision theory (SDT)[9] to prove that the these two models can be combined to infer neural dynamics in different cortical layers (laminar dynamics) using non-invasive MEG data. In general, SDT prescribes the optimal way of using quantitative tools to make statistical decisions in the face of uncertainty in the data[9]. This is often formulated in terms of decision rules. SDT has found applications in reinforcement learning[10] among other fields. Taking a statistical decision amounts to evaluating costs or losses based on same sample information combined with some other, e.g., prior or complementary, information. Here, we used SDT to reformulate compartmental and neural mass models as decision rules (besides other examples that mathematicians have considered as tools so far). Then, estimating neurobiological parameters of both compartmental and neural mass models is the same as making an optimal decision at different scales. After realizing this, we used insights from SDT to estimate biophysically accurate parameter sets that describe neural dynamics at both the macro- and microscales.

In refs. [11,12], we used a similar combination of computational models to analyse invasive animal data. However, we did not provide a mathematical proof of why such a combination can be considered and focused on invasive electrophysiology. Here we give a proof that establishes the mathematical basis of our approach. This also reveals limitations and suggests generalizations of our approach. Many alternative models can reproduce the same data (mean fields, neural masses, neural fields, etc.). The proof reveals which of them can be thought of as equivalent (the ones where parameters can be thought of as statistical decision rule estimates). It also suggests a similar approach for multimodal datasets (where models correspond to, e.g., functional magnetic resonance imaging and electroencephalography (EEG))[13].

Here, we also extend the work of Pinotsis et al.[11] from single subject animal data to multi-subject non-invasive human data. We analyse brain activity measured with MEG and consider between-subject differences in visually induced gamma oscillations. Our aim was to quantify the neurobiological mechanisms that underlie variability in human MEG data. The extension from animal to human data, together with the earlier work of Pinotsis et al.[11], suggest a two-step approach for understanding dynamics and neurobiology at the microscale using non-invasive electrophysiology. Step one: Construct a mean field model that includes the same neuronal populations as a validated compartmental model that captures biophysical properties of single neurons (e.g., the geometry of the dendritic tree, kinetics and densities of ion channels, inputs from subcortical areas) Fine tune its parameters to give similar predictions as the compartmental model. Test this with intracranial single subject recordings from non-human primates and rats. That was done in refs. [11,12]. Step two: Test the same mean field model with human data. This is presented here.

Below, we illustrate our approach using two independent human MEG datasets analysed earlier in refs. [6,14]. These datasets contain visually induced oscillations recorded during perception tasks. In earlier work, we had found that differences in the above oscillations between different people were due to differences in the level of inhibition in the cortical source. To validate our new approach, we asked whether we could confirm those earlier results (obtained using different computational models). We also asked whether our new model of laminar dynamics could identify the cortical layer where these differences were more pronounced. To address these questions we computed correlations between model parameters describing laminar dynamics (connections) and V1 size that is known to predict gamma peak frequency[15]. Interestingly, both datasets led to the same result. Differences in visually induced oscillations between different people originated from differences in the level of inhibition in deep cortical layers. This provides new insights into the computations performed by different cortical layers.

## Results

We used human MEG data to determine if we could explain individual differences in visually induced oscillations by localizing them to superficial versus deep cortical layers. We also used SDT to establish that the neural mass model we used to explain human MEG data could make the same laminar predictions as a compartmental model previously validated with animal data[11,12] (Supplementary Methods). Then we used hierarchical Bayesian modelling to explain individual differences in neurobiology and anatomy that underlie gamma oscillation variability.

**Computational models of brain dynamics at the micro- and macroscales predict similar laminar data.** First, we linked predictions of brain dynamics at different scales. This entailed a mapping between micro- and macroscale models that we discuss below. Following ref. [11], we here used a macroscale (neural mass) model whose parameters had been tuned to predict similar laminar data as a microscale (compartmental) model. (Similar has a precise, mathematically rigorous, meaning. Examples of similar predictions are shown in Fig. 1b and the mathematically rigorous meaning is explained in the Supplementary Methods.) This is based on SDT. Briefly, the macro- and microscale models are reformulated in terms of Bayesian decision rules. Then it is shown that a Bayesian observer can not distinguish between the data predicted separately by each model. Thus, they predict "similar" data. Alternatively, this means that if one fits both models to the same data using Bayesian inference then the model fits are the same (same error or accuracy). The model predicts the same peak frequencies for superficial and deep pyramidal cells and similar power distribution across frequencies (Fig. 1b) as the compartmental model available in the ModelDB database (https://senselab.med.yale.edu/modeldb/ShowModel.cshtml?model=136803&file=/JonesEtAl2009/mod_files/km.mod#tabs-2), including

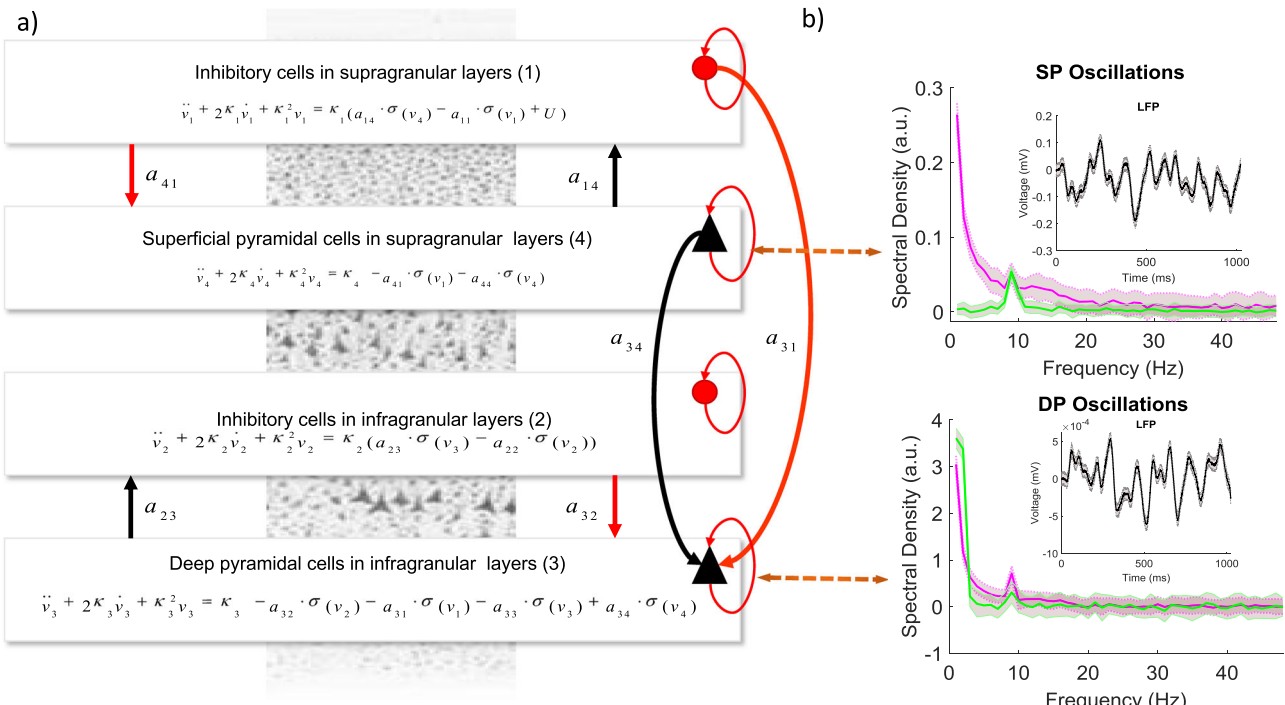

**Fig. 1 Model and predictions. a** Two pairs of excitatory (black) and inhibitory (red) populations occupy superficial and deep cortical layers. Firing rates within each population provide inputs to other populations and convolution of presynaptic activity produces postsynaptic depolarization. Arrows denote excitatory and inhibitory connections. All recurrent connections are inhibitory to preclude run-away excitation in the network. The same microcircuit was implemented both as a neural mass[11] and a compartmental model[5]. The equations describe the evolution of hidden states corresponding to activity in each of the four populations in the neural mass model. **b** Both models predict LFPs and power spectra. The top plot shows predicted power spectra from both models generated by superficial pyramidal neurons. Predictions from the neural mass model are shown in magenta, while predictions from the compartmental model are shown in green together with 95% confidence intervals. The bottom plot shows the same results for predicted spectra from deep pyramidal neurons. Insets in the top right corner of both plots show predicted LFPs. Note the peaked responses at 10 Hz that are reminiscent of spiking burst input that are also captured by the neural mass model responses.

alpha, beta and low gamma. It describes a similar distribution of spatiotemporal dynamics across layers (see below). To construct this model, we adapted the macroscale model of ref. [11] to deal with MEG as opposed to intracranial laminar data used in that earlier work. The new model predicts MEG oscillations from different cortical depths. Both macro- and microscale models describe the same microcircuit shown in Fig. 1a. This includes two pairs of excitatory–inhibitory (E–I) populations, one in superficial and the other in deep layers. Each E–I population pair is connected with intralaminar connections. Superficial and deep populations are also connected with interlaminar connections. Arrows correspond to excitatory (black) and inhibitory (red) connections. Figure 1a also includes the neural mass equations describing neural activity. A detailed description of the model can be found in ref. [11].

Besides describing the same microcircuit, the macro- and microscale models predict similar laminar dynamics. The mathematical proof of the equivalence of their predictions is based on SDT and is included in Supplementary Methods, see also ref. [16]. To make the two models predict similar dynamics, we used an analysis pipeline that combines existing and validated methodologies (Fig. 2) introduced in ref. [11]. In brief, this pipeline includes the following four steps: 1. Simulate data from the compartmental model. 2. Fit the mass model to these data. 3. Use the parameter estimates obtained as priors for fitting M/EEG data. 4. Obtain hidden parameters that describe laminar dynamics. We discuss them below. Here, this pipeline is also justified using theoretical arguments from SDT. This theory required that the microscale model be adapted to have the same number of parameters as the macroscale model. We changed the

compartmental model of ref. [5], by reducing the number of its connection parameters (considering all synaptic weights equal) and called the resulting model the symmetric compartmental model, see also refs. [11,12]. (In ref. [11], we found that evoked responses from the original model of ref. [5] and its symmetric variant were highly correlated, $r = 0.9343$, $p < 0.001$, see Supplementary Fig. S1A.) The remaining parameters are the weights of the connections between neural populations occupying different layers of the microcircuit shown in Fig. 1a. These are the same parameters that the macroscale model has. We then simulated power spectra from the microscale (symmetric compartmental) model (green data in Fig. 1b) and fitted the macroscale model to them using dynamic causal modelling (DCM) for steady-state responses[12,17,18]. This ensured the neural mass model has construct validity in relation to the compartmental model. DCM approach has been used to infer changes in synaptic plasticity in large cortical networks[17] in healthy and clinical populations[19] among other applications. It exploits the stationarity of variance of neuronal firing over the course of the task to efficiently fit neural mass models to MEG data. After fitting, we obtained the parameter values of the macroscale model that fitted best the predictions of the microscale model. The predictions from the macroscale model for these parameter values are shown in Fig. 1b in magenta. Predictions from both models are very similar. The correlation between these predictions was $r = 0.5$, $p < 10^{-3}$ for power spectra of superficial pyramidal cells and $r = 0.75$, $p < 10^{-4}$ for power spectra of deep pyramidal cells. The magenta and green curves overlap for frequencies above 3 Hz for deep layers (Fig. 1b, bottom) and 8 Hz for superficial layers (Fig. 1b, top). This difference in superficial layers is due to the fact

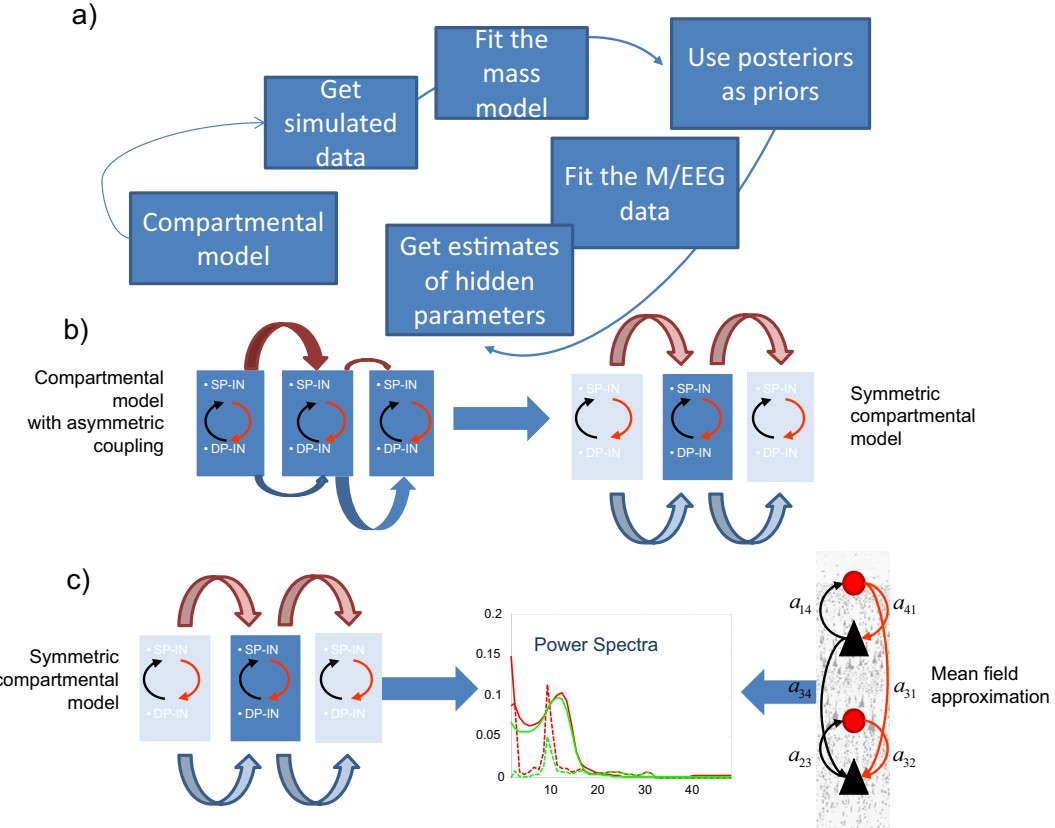

**Fig. 2 Outline. a** Schematic of our analysis pipeline. This summarizes the steps of our approach: 1. Simulate data from the compartmental model. 2. Fit the mass model to these data. 3. Use the parameter estimates obtained as priors for fitting M/EEG data. 4. Obtain hidden parameters that describe laminar dynamics. **b** Construction of the neural mass model: We first establish a similarity between the model of ref. [5] and its symmetric variant. Here horizontal arrows of different widths in the left panel denote asymmetric connectivities and delays between mini-columns depicted as rectangles containing Superficial and Deep Pyramidal cells (SP and DP) and Inhibitory Interneurons (IN). In the right panel, a symmetrization of the compartmental model reduces the number of connectivity parameters to be the same as those in a homologous neural mass model. **c** Construct validity of the mass model. To demonstrate this, we fitted the mass model to synthetic (laminar) data obtained from its compartmental homologue. This is justified by statistical decision theory. Red and green lines in the middle panel correspond to real data and model predictions. Solid and dashed lines to real and imaginary parts of crossspectra between deep and superficial pyramidal neurons.

| | Description | Max conductance (μS) Microscale model | Intrinsic connectivity Macroscale model (a.u.) |
|---|---|---|---|
| $a_{44}$ | SP → SP | 0.001 | 4.4 |
| $a_{14}$ | SP → SI | 0.003 | 4.8 |
| $a_{34}$ | SP → DP | 0.00025 | 23.3 |
| $a_{41}$ | SI → SP | 0.015 | 3.8 |
| $a_{31}$ | SI → DP | 0.0003 | 5.9 |
| $a_{11}$ | SI → SI | 0.0006 | 4.2 |
| $a_{33}$ | DP → DP | 0.005 | 2.2 |
| $a_{23}$ | DP → DI | 0.0003 | 4.6 |
| $a_{32}$ | DI → DP | 0.0075 | 6.9 |
| $a_{22}$ | DI → DI | 0.0006 | 4.16 |

**Table 1 Synaptic connectivity parameters for the microscale and macroscale models.**

that the macroscale model receives explicit white and pink noise input that propagates to all layers, while the microscopic models received Gaussian spike input targeting superficial layers, see refs. [7,12] for more details. Exogenous input to the compartmental model groups of 10 bursts (each consisting of 2 spikes separated by a 10 ms interval) with 100 ms intervals between the burst

groups. Despite these differences, we show below that SDT allows us to estimate neural mass model parameters for which the neural mass predicts the same neural dynamics as the compartmental model. This is done by fitting the neural mass to simulated data from the compartmental model.

To sum up, we adapted the microscale model of ref. [5] and fine tuned the parameters of a macroscale model describing the same microcircuit so that the two models make the predictions of the same frequency peaks and similar power distribution across frequencies in superficial and deep layers. The two models are thus functionally equivalent. The parameter values of the two models for which this happens are included in Table 1. Parameters for the microscale model are shown in the left column while the parameters of the macroscale model in the right. These include the connection parameters between the populations shown in Fig. 1: inhibitory interneurons in superficial layers, SI; pyramidal cells in superficial layers, SP; inhibitory interneurons in deep layers, DI; pyramidal cells in deep layers, DP. Each parameter corresponds to one arrow. There are in total ten connections in both the microscale and macroscale models.

In brief, coupled with a mapping from laminar dynamics to MEG sensors, the microscale model has been used to *simulate* MEG oscillations in different frequency bands[5,8]. Following our earlier work[6,14], we here used the fine-tuned macroscale model

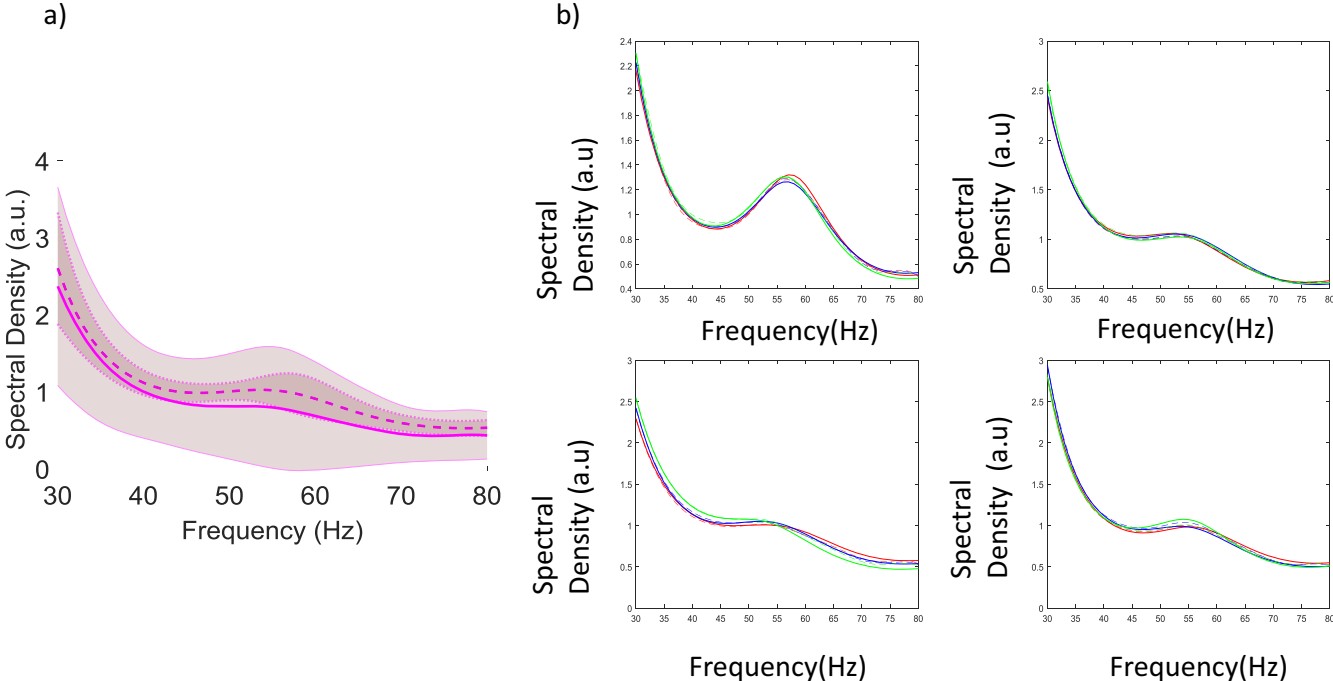

**Fig. 3 Model fits to data. a** Empirical responses (power spectra) and model fits are shown in dashed and solid lines along with 95% confidence intervals across subjects for the data from ref. [20] (smallest stimulus size; grand average across all subjects). These quantify variability across subjects and overlap for model predictions and data predictions. Differences in power spectra between conditions for the same subject are small and spectra corresponding to different stimulus sizes overlap. **b** Example model fits to individual subjects in data from ref. [16]. Fits are obtained for all three stimulus sizes simultaneously by modeling different sizes as condition-specific effects. The three stimulus sizes correspond to red, blue and green lines. Dashed and solid lines correspond to data and model.

and a similar mapping (given by $L_r(k, \varphi)$, see "Methods") to *fit* MEG data and explain individual variability in human brain oscillations. This is described below.

**Variability of visually induced gamma oscillations from different datasets reveals differences in the level of inhibition across different people.** We mapped differences in recorded power spectra from different people to differences in the macroscale model parameters and biophysics. We fitted the tuned macroscale model to oscillatory data from two different MEG datasets using DCM for steady-state responses[18]. Both datasets contained visually induced gamma oscillations reported in refs. [15,20]. These oscillations were recorded in different subjects. The spatial extent of the V1 cortical source in each subject varied. In ref. [20] this was the result of changing the size of the visual stimulus presented to the subjects. In ref. [15], this was measured with retinotopic mapping. The tuned parameters described above and in Fig. 2 were used as prior values for fitting the macroscale model to these MEG data.

We first fitted the macroscale model to the MEG data from ref. [20]. These included visually induced oscillations while 12 subjects viewed grating stimuli of three different sizes— $2^0$, $4^0$, and $8^0$. Thus the dataset from ref. [20] included oscillations induced by three different stimulus sizes. Grand averages of model fits across all subjects for the smallest stimulus size are shown in Fig. 3a. Dashed and solid lines correspond to data and model, respectively. Shaded regions include 95% confidence intervals. These quantify variability across subjects and overlap for model predictions and data. Model fits to *individual* subjects and conditions are shown in Fig. 3b. Fits are obtained after convergence of the EM algorithm[21]. They are tight: variance explained is above 95%. We fitted all three stimuli conditions simultaneously by modelling different sizes as condition-specific

effects, see ref. [14]. The three stimulus sizes correspond to red, blue and green lines. Dashed and solid lines correspond to data and model predictions. Differences in power spectra between conditions for the same subject are small and spectra corresponding to different stimulus sizes overlap. We fitted the data of each subject by changing the connection weights. These describe the strength of effective connections between different populations and describe the level of excitation and inhibition (E–I balance). Recall that priors for these parameters were chosen so that the macroscale model predicts the same data as the microscale model. After fitting individual subject data, the parameters of the macroscale model changed. We call the changes from priors, *connection changes*. Besides MEG oscillations, the two datasets from refs. [15,20] we analyse here, also included structural data, in particular V1 size, measured with MR spectroscopy. After obtaining subject-specific parameters (connection changes), we tested if the variability across subjects we observed in them is linked to V1 size and frequency and amplitude of MEG oscillations.

In Fig. 4a we included the correlations between connection changes and V1 size for the data reported in ref. [20]. We found four correlations with connection changes, three of them corresponding to changes in connections in *deep* cortical layers. The second pair of statistics in parentheses are partial correlations obtained when controlling for the rest of the connection changes. The parameters that showed significant correlations with V1 size were: changes in the drive to deep pyramidal neurons from superficial interneurons, $a_{31}$ (Fig. 4a, iii, $R = -0.664$, $p = 0.022$; $R = -0.858$, $p = 0.142$). Changes in the local drive to deep pyramidal neurons from other pyramidal neurons, $a_{33}$ (Fig. 4a, x, $R = -0.594$ $p = 0.046$; $R = -0.008$, $p = 0.992$). Finally, changes in the drive to deep inhibitory interneurons, $a_{34}$ (Fig. 4a, v, $R = 0.685$, $p = 0.017$; $R = 0.683$, $p = 0.317$). The only correlation

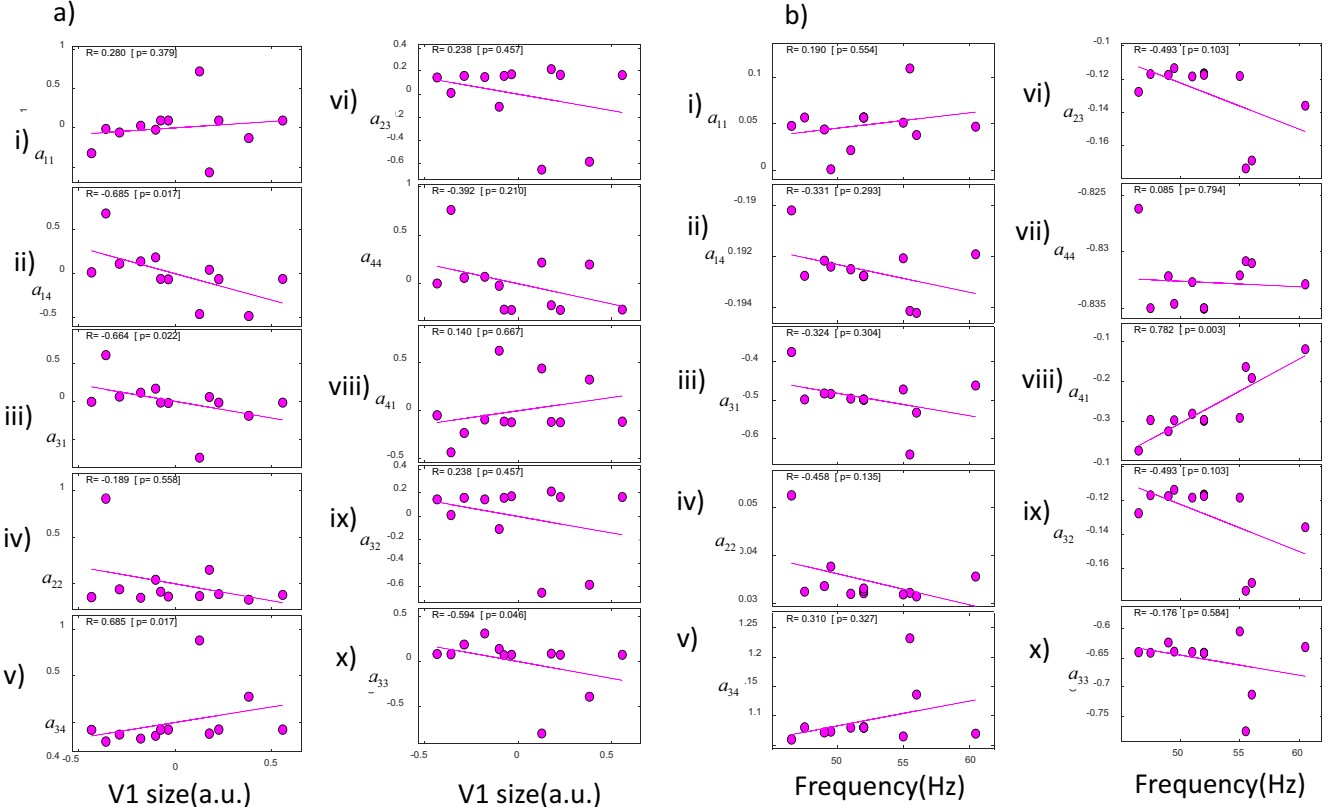

**Fig. 4 Connection changes in data from ref. [20]. a** Correlations between connectivity parameters and V1 size. Parameters were obtained by fitting data from ref. [20]. Changes in the drive to deep pyramidal neurons from both superficial interneurons, $a_{31}$ (iii) and deep pyramidals, $a_{33}$ (x) correlated with V1 size. Also changes in the drive to deep interneurons, $a_{34}$ (v) and drive to inhibitory interneurons in superficial cells, $a_{14}$ (ii). **b** Correlations between connectivity parameters and gamma peak frequency. Parameters were obtained by fitting data from ref. [20]. Changes to the drive to superficial pyramidal neurons correlated with peak frequency, $a_{41}$ (viii). Least-squares fitted line shown in magenta.

involving changes in *superficial* as opposed to deep connections was the drive to inhibitory interneurons in superficial cells from pyramidal neurons, $a_{14}$ (Fig. 4a ii, $R = -0.685$, $p = 0.017$; $R = -0.864$, $p = 0.136$). All above results did not remain significant when considering partial correlations or using Bonferroni correction for multiple tests at the $p < 0.05$ level. This was not surprising as our macroscale model predicts temporal differences in cortical dynamics between layers (vertically). Correlations with horizontal V1 surface are weak as the model does not describe interactions between neurons within the same layer of V1 (horizontally). This is because it does not describe the spatial deployment of cortical sources in detail. Spatial deployment can instead be described by an extension of neural masses known as neural fields, see our earlier work[14].

In Fig. 4b, we included the correlations between connection changes and gamma peak frequency. We found that, in this case, changes to the drive to superficial pyramidal neurons from inhibitory interneurons across individuals significantly correlated with peak frequency, $a_{41}$ (Fig. 4b, viii, $R = 0.782$, $p = 0.003$). Interestingly, this correlation remained significant under a Bonferroni correction for multiple tests. It was also positive. An increase in the inhibitory drive parameter in superficial layers led to an increase in the frequency peak. This positive correlation has also been found in other studies with similar models[6,22]. This is the reverse of the effect on the inhibitory time constant, whose increase leads to a peak decrease. This is thought to capture changes in IPSPs where an increase in inhibition leads to longer IPSPs and reduces the peak frequency[23]. Similarly to earlier results, this correlation did not remain significant when controlling for the remaining connection weights ($R = -0.758$,

$p = 0.452$). This was not surprising as the same change (increase or decrease) in gamma peak frequency (and amplitude) can result from increasing or decreasing many of the parameters of Table 1 (see ref. [6] for model parameter effects on gamma frequency and amplitude).

We also considered the correlations of connection changes with gamma amplitude (Supplementary Fig. S2B) and relative gamma amplitude change between conditions (three stimulus sizes; Supplementary Fig. S3). Gamma frequency, amplitude and relative amplitude change between conditions were the three data features that[20] used to quantify the gamma oscillation and response curve shape (while stimulus size increased). Thus, we considered the same features here. We found that changes in recurrent connections in *deep* inhibitory interneurons, $a_{22}$, correlated with changes in gamma amplitude across subjects (Fig. S3B, iv, $R = 0.608$, $p = 0.04$). No other connection changes were correlated. Similarly to the result above, $a_{22}$ did not remain significant under a Bonferroni correction and when controlling for the remaining connections ($R = 0.904$, $p = 0.096$). To sum up, six connection parameters correlated with V1 size and gamma oscillation features. Five of them were associated with deep neuronal populations.

We then fitted the macroscale model to data from ref. [15]. This was similar to our earlier work[6] where we had used a different macroscale model (neural field). The data from ref. [15] included visually induced oscillations from the visual cortex of 16 subjects recorded while subjects fixated on the centre of a screen and viewed a static, high-contrast, square-wave, vertical grating. Individual gamma peak frequency correlated with V1 size recorded with MR spectroscopy[15] (Supplementary Fig. S1B).

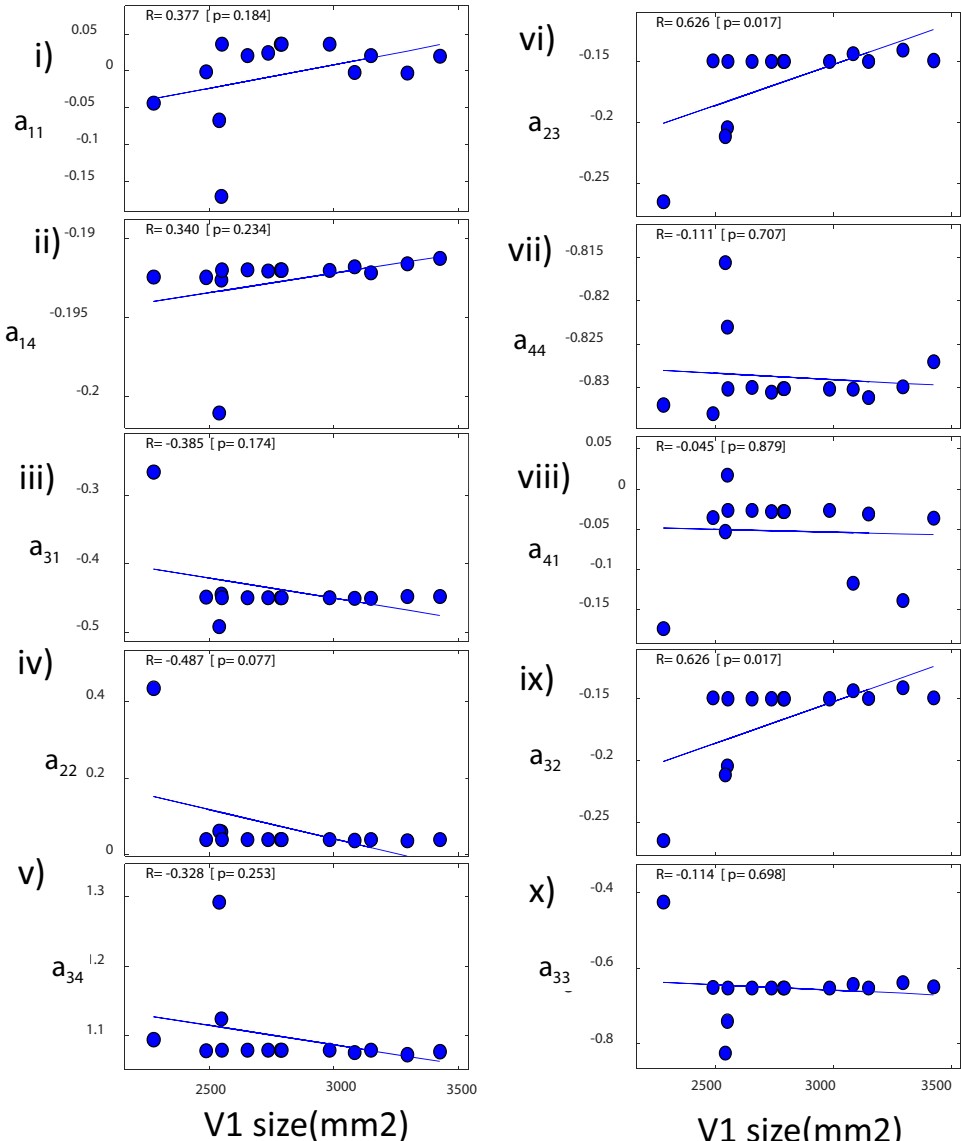

**Fig. 5 Connection changes in data from ref.[15].** Correlations between connectivity parameters and V1 size using data from ref.[15]. Changes in the connections between the *deep* pair of excitatory and inhibitory populations, $a_{23}$ and $a_{32}$, significantly correlated with different V1 sizes (vi and ix). Least-squares fitted line shown in blue.

Connection changes are shown, for each subject, in Fig. 5 and Supplementary Fig. S2A. Figure 5 which shows correlations of connection changes with V1 size. The horizontal axes show V1 size in mm$^2$. We found that changes in the connections between the *deep* pair of excitatory and inhibitory populations, $a_{23}$ and $a_{32}$, significantly correlated with different V1 size. (Fig. 5vi and ix, $R = 0.626$, $p = 0.017$; note that only changes from priors between different subjects were the same for the two parameters. The parameters themselves were *not* the same as their priors were different.) This did not remain significant with a Bonferroni correction and when controlling for the remaining connections ($R = 0.251$, $p = 0.684$). We also considered the correlations of connection changes with gamma peak frequency (Supplementary Fig. S2A). No changes were significantly correlated. Also, $a_{41}$ did not correlate with peak frequency in this dataset.

In brief, in both datasets, we found six parameters that were correlated with V1 size, five of them relating to connections targeting *deep* cortical layers. Besides V1 size, we also found significant correlations with gamma peak frequency and amplitude in one dataset. Taken together, we found correlations

between eight connection changes and V1 size or gamma oscillations. Six of them described differences in the level of *inhibition* across subjects. In the visually induced oscillations from 12 subjects reported in ref.[20] that we analysed, we found changes in the excitability of inhibitory interneurons. In the oscillations from 16 subjects reported in ref.[15] that we also analysed, we found differences in the drive and input to inhibitory interneurons. In both cases, the interneurons involved are those appearing in deep cortical layers of the microcircuit of Fig. 1.

To sum up, the above results suggest that variability in MEG oscillations across subjects originated from differences in the level of inhibition in deep cortical layers and correlated with V1 size. However, all but one correlation did not remain significant when corrected for multiple comparisons using a Bonferroni correction or accounting for other sources of variance using partial correlations. Previous work with a similar model had found correlations between parameters and gamma oscillation features that survived multiple comparisons[22]. This is interesting and could be explained by the fact that prior parameters in that earlier

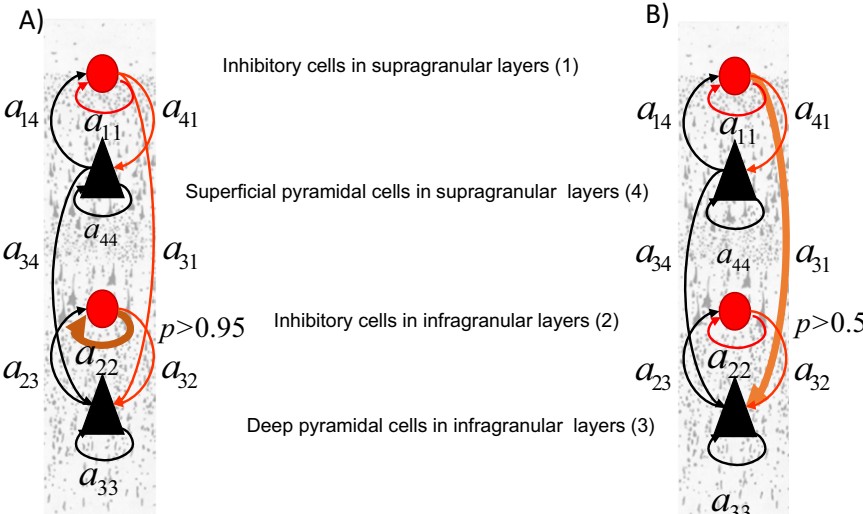

**Fig. 6 V1 size predictors. a** We scored alternative GLMs where predictors of variability in V1 included any combination of the connections (arrows) in Fig. 1a. We found that for the data from ref. [15] V1 size could be best predicted by the recurrent connectivity of deep inhibitory interneurons, $a_{22}$ (brown arrow). Evidence in favour of a GLM including $a_{22}$ was very strong $p > 0.95$. **b** Same as in **a** for data from ref. [20]. V1 size variability reported in ref. [20] could be best predicted by the inhibitory drive to deep pyramidal cells, $a_{31}$ (brown arrow). Evidence for the corresponding GLM was weak $p > 0.5$.

work did not include information about differences in neural activity between different cortical layers. This rendered that earlier model more flexible while explaining MEG data. Constraining parameters to differentiate between superficial and deep layer activity like we did here poses an extra challenge given that information about laminar (depth) differences in neural activity might be limited in MEG data. To test our hypothesis about individual variability in MEG oscillations originating from deep cortical layers, we turned to hypothesis testing using Bayesian methods. This has the advantage that it allows us to test *all* alternative hypotheses in terms of which model parameters drive individual variability. This also addresses the above problem of correlations not remaining significant by expressing significance in terms of a (Bayesian) odds ratio (known as Bayes Factor (BF)). Specifically, we used a recent approach known as parametric empirical Bayes (PEB) (see Methods and Supplementary Methods for a summary of the theory behind this[24,25]). This has been used to understand individual differences in effective connectivity in large cohorts of healthy people and patients[14,24–26]. This approach provided a BF that quantifies the relative probability of a general linear model (GLM) that included the connection change as a predictor of a data feature (e.g. V1 size) vs. a GLM that did *not* include this parameter. If, e.g., BF > 20:1 we can conclude that this connection change predicts the chosen data feature with probability $p > 0.95$ (ref. [27]). This is discussed next.

**Individual differences in MEG oscillations reflect differences in V1 size and deep layer activity**. We considered local intracortical connectivity between E–I neurons and asked whether connection changes predict different data features: V1 size and gamma peak frequency. This provided an answer to the question which connections and cortical layer might drive differences in these data features. We used the PEB approach. It allowed us to test *all* alternative possible combinations of connection changes that could predict data features. These correspond to combinations of the ten connections shown in Fig. 1a (arrows). The connection changes were predictors that entered a GLM. The PEB approach can be thought of as a numerically efficient way to use Bayesian statistics to score alternative GLMs. Because it is Bayesian, it downweighs the contribution of outliers (quantified in terms of

posterior variance). Here we considered alternative GLMs that did not include one or more of the connections between populations shown in Fig. 1.

We computed the BF between a GLM model that included a particular connection change as predictor vs. a GLM without changes. This quantifies the probability of a particular connection predicting the data features. We found that variability in V1 size reported in ref. [15] could be best predicted by the recurrent connectivity of deep inhibitory interneurons, $a_{22}$ (Fig. 6a; very strong evidence, $p > 0.95$), while variability in V1 size reported in ref. [20] could be best predicted by the inhibitory drive to deep pyramidal cells, $a_{31}$ (Fig. 6b; weak evidence, $p > 0.5$). Gamma peak frequency was not predicted by connection changes, like $a_{41}$ that is known to underlie the PING mechanism[28]. Thus, our result suggests that variability in the level of inhibition in deep cortical layers obtained by fitting MEG spectra predicts V1 size. To sum up, differences in oscillations across people seem to reflect macroscopic differences in V1 size and microscopic differences in neuromodulation specific to deep layers 5/6.

**Discussion**

We found that individual differences in visually induced gamma oscillations recorded with MEG reflect differences in the level of inhibition. Our results fit with studies that found GABAergic channels to play a prominent role in generating gamma oscillations[29] and that gamma oscillation variability reflects differences resting in GABA concentration measured with MR spectroscopy[30]. This follows a long line of work where PING and similar circuit mechanisms have been used to explain the generation of gamma oscillations[31–33]. Also, they confirm results we obtained earlier using different models and statistical analyses. In ref. [6] we found that individual differences in gamma oscillations reflected the level of inhibition in the cortical source. They were driven by variations in the excitatory drive to GABAergic interneurons. In ref. [14], we also found that these differences were driven by variations in not only the drive but also interneuron output.

In the visual cortex interneurons spread across layers[34]. Thus, the earlier work described above did not answer the question if there are any differences in the level of inhibition between cortical layers. Here, we used a combination of macro- and microscale computational models to address this question. Both models

predicted the same laminar data, that is, responses from different cortical depths. By fitting the macroscale model to MEG data, we were able to locate the cortical layer that might be the origin of differences between oscillatory responses of different subjects. We found that the variability in the level of inhibition across subjects was specific to *deep* cortical layers. Thus, our results suggest that differences in gamma oscillations between subjects originate from deep cortical layers. In future work, we will ask whether this holds in other areas beyond V1.

Controlling for multiple comparisons and the effect of remaining variables, most correlations did not remain significant. This is not surprising as we used non-invasive data to fit the model. However, three further arguments offer evidence in support of our result that deep layers account for differences in gamma oscillations.

First: animal data. Deep layer beta oscillations control superficial gamma oscillations[35]. In other words, interactions between deep cortical inhibition and gamma oscillations of the sort we found here using non-invasive *human* brain data have also been observed using laminar electrodes in *animals*. A recent study[35] found that during memory delay, activity in deep cortical layers releases inhibition that targets superficial cortical layers. This interaction is expressed via alpha/beta oscillations that are known to inhibit distracting stimuli[36]. Neural activity in layers 5/6 might modulate activity in layers 2/3 and from there to layer 4 input[37]. This suggests a control of superficial gamma oscillations from deep alpha/beta inhibition. During delay, inhibition release allows recurrent excitation in superficial layers that allows memory ensembles to persist[38]. The balance between gamma and lower (alpha/beta) frequencies has also been shown to reflect volitional control of working memory originating from frontal areas[39].

Second: a greedy search approach. We tested all possible combinations of model parameters obtained by fitting gamma oscillations and showed that parameters associated with deep populations could predict V1 size that correlated with gamma peaks[15] (see also Supplementary Fig. S1B).

Third: Predictive Coding (PC). PC suggests that deep layer activity corresponds to different hypotheses about sensory input that different people have[17,40]. Deep layers are thought to represent expectations of sensory input. Different expectations can, in turn, be due to different prior experiences. Several studies have suggested this. A recent study using invasive, laminar electrodes in monkeys found that changes in deep cortical inhibition correlated with stimulus predictability, or how confident the monkey was about the upcoming stimulus[41]. This result can be reconciled with our current results in the context of the theory of PC: changes in deep inhibition can reflect both (1) different processing of the same sensory input across people who differ in their predictions that we found here and (2) different sensory inputs (with different predictability) the same subject has, which was found in ref. [41].

In brief, animal studies found that deep layer activity in sensory areas modulates superficial gamma oscillations and is the result of feedback from higher areas[35,39]. Here, we found that deep cortical inhibition is also related to gamma oscillation variability observed in human subjects using correlations and a greedy search. Specifically, we found that deep cortical inhibition parameters correlated with V1 size. This, in turn, was shown to correlate with gamma oscillation variability[15]. This result also fits with PC: differences in modulation of superficial gamma by deep layers found using invasive recordings express different levels of predictability[41].

Our approach focuses on explaining differences in neural dynamics between different cortical layers. Because of constraints imposed upon the parameters of the microscale (compartmental) model, spatial effects within the same layer were neglected (e.g.

interactions between cortical columns and the dependence of gamma peak frequency on the horizontal spread of the underlying cortical source). In future work, we will study the relationship between population activity predicted by the neural mass (macroscale) model and the symmetric compartmental (microscale) model we used here. Spatial effects could be addressed by extending our approach to a different class of mean field models besides the neural masses considered here, called neural fields[14]. In earlier work, we used neural fields to explain the relation between spatial and biophysical properties of cortical sources and observed brain dynamics[6]. We also revealed differences in brain structure (V1 size) and inhibitory function between different people[14].

We here used combinations of macroscale and microscale models, not isolated models. We estimated neural activity in different cortical depths of the visual cortex using non-invasive, human MEG data. We suggested that isolated models spanning a single spatial scale are not sufficient to fully understand such data and instead model combinations should be used. This allowed us to obtain information about human cortical layer activity and anatomy and exploit information at different spatial scales simultaneously.

More generally, our approach can help address the heterogeneity observed in neurological diseases and disorders. The link between deep layer activity, cortical inhibition and gamma oscillations found here and in refs. [35,41] is relevant to such disorders, where the excitation to inhibition (E–I) balance is disrupted[42]. Individual differences in E–I balance are one prominent feature of heterogeneity observed in neurological disorders[43]. Based on our results, we predict that E–I balance disruption will be more prominent in deep cortical layers. In future work, we will test this hypothesis More generally, obtaining connection parameter estimates informed by laminar dynamics allows one to consider several applications of DCM models where neural dynamics show interesting modulations depending on E–I balance, including circadian dynamics, working memory, neurological disorders, etc., see e.g. refs. [18,44–47].

Heterogeneity of neurological diseases and disorders is studied using non-invasive MEG data that sample brain responses at the macroscale. At the same time, it is expressed in the microscale (cortical circuit level) as, e.g., differences in the E–I balance. Thus, we need *multiscale* approaches to study this heterogeneity. Using isolated models is not enough. Combinations of models spanning the macro- and microscales should be considered. Spatial information at the macroscale is accessible via non-invasive human brain imaging. Spatial information at the microscale and the structure and function of local cortical circuits is accessible in vitro or through invasive recordings in animals. Thus to exploit information at both scales we need hybrid models of the sort considered here.

To sum up, our multiscale approach enables the study of E–I balance differences at the cortical circuit level using M/EEG data. It opens up the way to testing hypotheses about microscale cortical circuit structure and function in human patient populations.

## Methods

**Data**. We used two datasets to test whether our model yields similar results in datasets taken from similar but independent experiments. First, we used visually induced oscillations taken from ref. [20], where whole-head MEG recordings were acquired using a similar CTF radial gradiometer system as above sampled at 1200 Hz. In that task, a time series of the response at the location of the peak in each trial was generated by spatially filtering the sensor-level data through the corresponding beamformer weights at that location. These "virtual sensor" time series were then used to obtain power spectra from 12 healthy subjects while they viewed grating stimuli of three different sizes— $2^0$, $4^0$ and $8^0$. In the second dataset, we used MEG beamformed data from ref. [15]. Using virtual sensor or beamformed data allowed us to make inferences pertaining to neural activity (not the observation model). These data included visually induced oscillations from the visual cortex of 16 subjects.

In that task, subjects fixated on the centre of a screen and viewed a static, high-contrast, square-wave, vertical grating. MEG data were recorded using a whole-head CTF axial gradiometer system with 275 channels, sampled at 600 Hz. Three electrical coils were placed at fiducial locations and used to monitor subject head movement. Data were analysed using SPM8 (http://www.fil.ion.ucl.ac.uk/spm). We used an LCMV beamformer algorithm implemented in SPM8 to quantify source power in the time window between 0.5 and 1.5 s after stimulus onset relative to baseline power over one second preceding stimulus onset. We located peak gamma activity in the medial occipital cortex, and at this peak location we used the beamformer weights to extract the time series of the virtual sensor. For more details on the above two datasets, see refs. [15,20].

**Biophysical models.** To construct the microscopic model (that we called symmetric compartmental model), we adapted NEURON code from ref. [5]. The original model of ref. [5] included a compartmental model of the local microcircuit shown in Fig. 1a. This comprised pyramidal neurons and interneurons[48]. This model was later extended to a network model of a cortical column in a key paper by Jones et al.[5]. The resulting network model provides detailed descriptions of intracellular (longitudinal) currents within the long apical dendrites of synchronized cortical pyramidal cells. It describes neuronal morphology and laminar structure of a cortical column and characterizes cellular and circuit level processes measured with multielectrode arrays or MEG. Driven with Gaussian input in the time domain, the model accurately reproduced the S1-evoked response to a tap on the hand and described the intracellular currents that give rise to signal polarity. We here used a variant of this model where we increased the number of inhibitory units from 3 to 10 per layer, so that their number was equal to the number of the principal cells within each mini-column. The original model comprised 10 PNs in layers 2/3, 10 PNs in layer 5, and 3 INs in both layers. The synaptic architecture followed general tenets of cortical micro-circuitry where FF connections target the granular layer and FB connections target agranular layers, see e.g. ref. [6] for a further discussion. Modelling of single neuron morphology and physiology followed[48], using the same parameters as in ref. [5]. Details can be found in ref. [11]. To ensure that relative differences in interneuron densities were accommodated, we multiplied the maximum conductance values of the corresponding connections by a factor of 0.3. To reduce the number of connection parameters (synaptic strengths) to be equal to those of the macroscopic model, we assumed that the connection weights between different mini-columns comprising the microcircuit shown in Fig. 1a were the same. This assumed symmetry constraints on horizontal connectivity (within each cortical layer) of the sort assumed in mean field models that describe aggregate activity over hundreds of neurons. We then simulated data from the symmetric compartmental model and made them amenable to a further DCM analysis *sim-Data.mat*. The model was integrated using the implicit functionality of NEURON. We then used a a simple Welch method for obtaining spectral density estimates. This was also used in DCM.

We also implemented a macroscale (neural mass) model that describes the same local cortical circuit as the microscale model above. This assumes symmetry constraints on horizontal connectivity (within each cortical layer) of the sort assumed in neural mass models that describe aggregate activity over hundreds of neurons. It is implemented as part of SPM software *spm_fx_cmc_BS.m*. Then we fitted this model to the simulated data above using DCM and estimated its parameters as described in the next section.

**Parameter estimation.** To fit the biophysical neural mass model to simulated and MEG data we used DCM for steady-state responses[18], that is implemented in the function *spm_dcm_csd.m* of the DCM toolbox. The inversion of neural mass models above uses the standard DCM approach that we summarize below. First, it assumes that the neural mass model is driven by endogenous neuronal fluctuations that produce observed power spectra according to the following likelihood model, see also[14]:

$$
\begin{aligned}
g(\omega) &= g_Y(\omega, \theta^{(1)}) + g_N(\omega) + \varepsilon^{(1)} \\
g_Y(\omega, \theta^{(1)}) &= \sum_k \left| QL_r(k, \varphi) T(k, \omega, \theta^{(1)}) g_u(k, \omega) \right|^2 \\
g_n(\omega) &= \alpha_n + \beta_n/\omega \\
g_u(\omega) &= \alpha_u + \beta_u/\omega \\
Re(\varepsilon^{(1)}) &\sim \mathcal{N}(0, \Sigma(\omega, \lambda)) Im(\varepsilon^{(1)}) \sim \mathcal{N}(0, \Sigma(\omega, \lambda)).
\end{aligned}
\tag{1}
$$

Here, $g_u(\omega)$ is a spatiotemporal representation of fluctuations or inputs driving induced responses, which we assume to be spatially white and a mixture of white and pink temporal components, $L_r(k, \omega)$ is the Fourier transform of the lead field of the MEG virtual sensor and $Q = [q_1, q_2, q_3, q_4]$ is a vector of coefficients that weights the contributions of each neuronal population (among the 4 shown in Fig. 1a) to the observed MEG signal. These contributions are based on differences in anatomical properties and the lead field configuration of each population (e.g. inhibitory neurons do not generate a large dipole[49]), where each electrode or sensor has its own sensitivity profile. The transfer function $T(k, \omega, \theta^{(1)})$ are the Fourier transform of the impulse response or first-order Volterra kernel associated with the ordinary differential equations shown in the boxes of Fig. 1a. This transfer function describes how each of the four populations response to neuronal fluctuations, where the model parameters describe the connectivity architecture mediating

responses, the observation function $\varphi \subset \theta^{(1)}$ and the spectra of the inputs and channel noise, $\{\alpha_n, \alpha_u, \beta_n, \beta_u\} \subset \theta^{(1)}$.

In summary, Eq. (1) expresses the data features $g(\omega)$ as a mixture of predictions and sampling errors $\varepsilon^{(1)}$ with covariance $\Sigma(\omega, \lambda)$. Gaussian assumptions about these sampling errors provide the likelihood model at the first (within subject) level: $p(g(\omega)|\theta^{(1)})$. The predictions are themselves a mixture of predicted cross spectra and channel noise $g_N(\omega)$. This concludes the description of the likelihood model.

The optimization of this likelihood model is based on optimizing a free energy bound on the model log-evidence. The free energy bound is optimized with respect to a variational density $q(\theta) \sim \mathcal{N}(\mu, C)$ on the unknown model parameters. By construction, the free energy bound ensures that when the variational density maximizes free energy, it approximates the true posterior density over parameters, $q(\theta) \approx p(\theta|g_Y(\omega)_i, m)$. This bound is given by the Free Energy, see[6] for more details.

Using DCM, we fitted the neural mass model to power spectra between 30 and 80 Hz from different subjects reported in refs. [15,20]. This frequency range contains gamma frequencies captured with MEG[15,20]. This analysis was similar to our earlier work described in refs. [6,14] where a detailed description of the DCM procedure is given. To fit data from ref. [20], we fitted all three stimuli conditions (different stimulus size described above and in ref. [20]) simultaneously by modelling different sizes as condition-specific effects (the $B$ matrix in DCM), see ref. [14]. We next consider how this likelihood model is placed within a hierarchical model of responses from multiple subjects.

**Between-subject differences.** To model between-subject effects we used a hierarchical Bayesian inference approach known as PEB, see ref. [14] and the function *spm_dcm_peb_bmc.m* in the DCM toolbox included in SPM. At the first level, we used the neural mass model described above and in "Results". At the second level, we modelled individual differences in gamma peak frequency and amplitude. These were used as regressors in the design matrix entering the GLM at the second level of a hierarchical Bayesian model.

Below we summarize the PEB approach. This uses a hierarchical model to include constraints on the posterior density over model parameters provided by the level above. In variational Bayesian inference, the approximate posterior over the second level parameters is obtained by optimizing its sufficient statistics (i.e., mean and covariance) with respect to a (second level) free energy, see refs. [12,14,24,25] for more details. This hierarchical model accommodates both within- and between-subject effects given by the following equations:

$$
\begin{aligned}
y_i &= \Gamma_i(\theta^{(1)}) + \varepsilon_i^{(1)}, \\
\theta^{(1)} &= \Gamma(\theta^{(2)}) + \varepsilon^{(2)}, \\
\theta^{(2)} &= \eta + \varepsilon^{(3)},
\end{aligned}
\tag{2}
$$

where $y_i$ is a matrix of $i$th subject responses, $\Gamma_i(\theta^{(1)}) = g_Y(\omega, \mu, \theta^{(1)})$ represents the (differential equation or dynamic causal) model that generates these responses with parameters $\theta^{(1)}$ and $\Gamma\theta^{(2)}$ is the between-subject (second level) model that describes intersubject variability in the parameters of the first-level model. The second-level maps second- to first-level parameters (e.g., group means to subject-specific parameters), where $\varepsilon^{(1)}$ represent random effects at each level (e.g., intersubject variability and observation noise).

In the context of non-invasive electrophysiology, the hierarchical model (3) poses the difficult inversion problem of finding neural source estimates in the context of intersubject variability. This involves (i) partitioning the covariance of observed data into observation error and components that can be explained in terms of neuronal responses that themselves entail components due to second level (between-subject) variability; (ii) exploiting differential equation models to provide anatomical and physiological constraints on the explanation for first-level (within subject) responses—usually in terms of (synaptic) connectivity estimates. The second-level covariance components specify whether the parameters of the dynamical model at the first level are random or fixed effects, while dynamical models provide predictions of the dynamics at source and sensor space, which depend upon cortical anatomy and physiology.

In summary, hierarchical or empirical Bayesian modelling of the sort implied by Eq. (3) allows us to perform efficient source reconstruction and obtain connectivity estimates by replacing phenomenological constraints (e.g., based on autoregressive modelling and temporal smoothness considerations) by spatiotemporal constraints based on models of neuronal activity. This can be thought as an alternative to autoregressive modelling, which model statistical dependencies among measured signals—as opposed to the neuronal processes generating measurements. In DCM, one uses a forward or generative model of distributed processing to estimate the (coupling) parameters of that model. Inference then proceeds assuming nonlinear within-subject effects and linear between-subject effects. This allows one to distinguish among competing hypotheses about the mechanisms and architectures generating the data and the nature of group effects in multiple subject studies.

The key thing about the free energy is that it can be evaluated (using Bayesian Model Reduction; BMR[25]) without optimizing the first-level posterior. This means the second-level parameters (e.g., group means) can be optimized or estimated, for any given model of priors, without reinverting the model at the first level. Technically, the inversion of the hierarchical or empirical Bayesian model only requires the posterior density from the inversion of each subject's DCM. In short,

the use of BMR allows one to make inferences at the group level without having to re-estimate subject-specific parameters; see ref. [20] for details and a study of robustness of this scheme.

**Statistics and reproducibility**. Significant test was conducted using Spearman's Rho implemented in Matlab (http://mathworks.com), and *p* values were corrected for multiple comparisons using the Bonferroni correction. Our results can be reproduced using code available at https://github.com/pinotsislab/MicroMacro/ and using the DCM toolbox of SPM12, https://www.fil.ion.ucl.ac.uk/spm/software/spm12/.

**Reporting summary**. Further information on research design is available in the Nature Research Reporting Summary linked to this article.

## Data availability

The data that support the findings of this study are available from the corresponding author upon reasonable request.

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

## Acknowledgements
This work was supported by UKRI ES/T01279X/1, NIMH R37MH087027, ONR MURI N00014-16-1-2832 and The MIT Picower Institute Innovation Fund. We thank Dr. Daniel Gibson for a careful reading of the manuscript and important suggestions, Dr. Sam Schwarzkopf for illuminating discussions and providing data and Professor Krish Singh and Dr Gavin Perry for providing data.

## Author contributions
D.A.P. conceived the study and carried out the data analysis. E.K.M. provided critical feedback and helped shape the research, analysis and manuscript.

## Competing interests
The authors declare no competing interests.
