## [Peer Review File · Communications Biology]

Reviewers' comments:

Reviewer #1 (Remarks to the Author):

Under the scope of individual differences in clinical heterogeneity, this article proposes the importance of advances in the neurocomputational model. Under the significant necessity of understanding of individual differences, this article aimed to deepen the knowledge of the variations on cortical gamma-band oscillation during visual perception. In previous works, it has been known that individual variations in the gamma oscillation were due to regional GABAergic concentration and the level of inhibitory inputs in the sources. It has been found that cortical inhibitory activity may contribute to the gamma oscillation. However, it is still not clear as to whether different layers (superficial and deep layers) at the macro- and micro-scale may play differential roles on the individual differences in the gamma oscillation. In this article, the authors modelled layered architecture-based estimation method based on non-invasively recorded (MEG) signals and fitted their model to the superficial and deep layers. Based on the observations, authors conclude that deep layer, but not superficial layer, of “both” macro and micro scale models, were the sources of the individual differences. Besides, it is claimed that combined modelling of both macro- and micro-scale model is necessary that was not sufficient with an isolated model.

The intent of the paper is quite essential to the applied neuroscience field. It is also worth giving credit the message that deepening the analytical model by incorporating macro- and micro-scale at different cortical layers would advance the field of research. The method section seemed to be well-written and relatively easy to understand. In the meanwhile, the introduction and discussion were crude, and it was hard to read and follow the intent, lacking detailed explanation as to why the authors interpret the results. The set-up of the neural mass (macroscale) and microscale; these terms were used intermixed; it was a bit confusing. What was exactly how “macro” it was (see below). There are many places I have questions on, which can be described in detail, without skipping only to the main points. Also, the figures and the figure captions may be improved for the sake of ease of reading. They have re-analyzed two independent existing MEG data, while participants viewed visual stimuli. Induced power spectra data of 0.5–1.5 sec after stimulus onset were analyzed at the “virtual” sensor. But it was not clear why the virtual sensor conversion was necessary. Given the results that the spectra pattern observed between the macro- and micro-scale were the “similar”. Yet it was not clear to me what it means in terms of cortical activity with respect to the “individual differences”, which was meant to be the purpose of the article. Authors also mention about the relation to cortical volume in the V1 area as well as the E-I ratio that has been already proved by previous research. I understand varieties of the aspect of neural markers are interrelated; however, I had kept distracted what was the main finding and message in this article. In addition, in the discussion, the authors proposes the future research will investigate E-I (excitatory-inhibitory) ratio. Yet given the current form of flows of their evidence and logic, they describe too far to mention the E-I ratio; such speculation need to be toned down as no direct evidence had been analyzed here on E-I. The results seem to support the authors claim; however, the way it is written was too simple. The authors may have some ideas and logic as to why the current results are so related to the individual differences in the E-I ratio observed in GABA, the quantity of evidence is lacking to support the speculation. Finally, the authors first propose an intent of developing a novel computational model to investigate the heterogeneity in clinical research populations; there are little connection I could find in these analyses and results. The most troublesome (for me to understand) was the 2x2 design of the depth of layers (superficial and deep) and scales (macro and

micro). Notably, later in the text, the macro is expressed as lateral connection (should I interpret it as horizontal?). The term for macroscale, I would have expected more remotely interconnected network-level connection, whereas the micro is more regional and laminar (within a column). I think this could have been stated clearly from the beginning. Thus, the analysis was in the end, focusing only on the two columns in the visual cortex, which is very regional to me. Authors could have stated that the validation was solely focusing only on the visual cortex, and this result may not necessarily generalize to the other regions or a network-level, etc. To summarize my impression, all authors could re-write and reconsider the flows in the main introduction in a lay term as possible with enough background logic and explanations. Some changes in the figures may help readers to understand better.

Below I note “subsets”, not all, of questions what I had to stop and think, read again back and forth, trying to understand what authors wanted to claim. There are too many sentences where I had to stop reading. If I need to evaluate this article again, I hope the future version may be easy to read flawlessly so that a reviewer can fully engage in the judgment of the contents of the results better.

Page2, a paragraph starting with “Here, we also extend the work of [8]...”.

The description seems too simple, like a reading a list of brainstormed ideas. Authors could elaborate more on details.

Page2, a paragraph starting with “Below, we demonstrate our approach using”.

A line, “we demonstrate our approach using two ...” It is awkward for me. Please rephrase.

A line, “we had found that differences in these oscillations...” Please be specific to what the “these” refers to.

A line, “To validate our new approach, we asked whether...” is stating the same meaning as the very first line of this paragraph. It is overlapping, so it may not be necessary.

A line, “... the cortical layer where these differences were more pronounced.” Please specify where was more pronounced?

Page 3 Results section,

The last sentence of the first paragraph, a sentence “...individual differences in single subject neurobiology...” What does it mean the “individual differences in a single subject”?

In the footer, 1, “Similar” also has a precise, mathematically rigorous, meaning. Where is the “1” on this page? I found the same sentence was described in the Methods.

Page 4,

A line, “Second, we simulated power spectra from the microscale model.” Please explain why you performed this. Please provide more explanation of why it needs to be done.

A line, “Predictions from both models are very similar”. Please provide statistical, quantitative proof.

A line, “The magenta and green curves overlap for frequencies above ... This difference in superficial layers is due to...” Why talked on an overlap and then all of the sudden talks about a difference?

The last line of the paragraph, “This difference in superficial layers is due to the fact that ...” where can I find the fact? Please provide a point of reference. I would appreciate a more detailed explanation here.

A line, “... the same frequency peaks and similar power distribution across frequencies in superficial

and deep layers.” It was not clear to me why this awkward way to analysis was needed by adapting the microscale model and a macroscale model that is describing the same microcircuit... please provide logic for why adapting one to the other was necessary.

A title, “Variability of visually induced gamma oscillations from difference datasets...” Please be specific on “datasets”. May it be better to replace “datasets” with “visual stimuli on different observers”?

Page 8, In Discussion,

A line, “we found that individual differences in gamma oscillations reflected the level of inhibition in the cortical source.” Does it mean, this article simply replicated the previous study, [6]? I do not see much differences in the sentences between the very first sentence of the same paragraph (“We found that individual differences in visually induced ...”)and this line.

In the second paragraph, “subjects’ responses originate”. Does it mean subjects’ neural responses originate? I hardly believe subjects’ physical response (like pressing buttons) originate from the visual cortex. Please be specific.

Page 9.

In a paragraph starting “Third: ...”

A line “1) difference processing of the same sensory input across people who differ in their predictions that ...” In this paragraph, describing a link to predictive coding theories. I was not entirely convinced on this line. Please provide more explanations.

A line, “Here, we found that deep cortical inhibitions are also related to gamma oscillation variability ...” Again, I am not convinced how are they related. Please provide more explanations.

A line, “We also revealed differences in brain structure and function between different people.” I was not sure if any structure (grey matter volume) was related to the function in this article?

Page10

A line, “At the same time, it is expressed in the microscale as e.g. differences in the E-I balance. Thus, we need multiscale approaches to study...” I did not understand why we need a multiscale approach at all. Please describe. Same for the line, “Using isolated models is not enough”. Please kindly refer to where I can find the evidence that the isolated model was not enough. If not enough, how much does it lack compared to the multiscale approach, where I can see the quantitative results?

A line, “Spatial information at the macroscale is accessible via non-invasive human brain imaging. Spatial information at the microscale and the structure and function of local cortical circuits is accessible in vitro or through invasive recordings in animals.” I understand that proposal of a combination may help translate animal data at the microscale to macro-level human or network-level data. However, to be simple, collection of both data from the same individual in vivo may provide the best result rather than combining animal data to a human directly. Please provide more logic here to convince readers. Just stating “combination should be considered” without proof may not be enough to persuade readers fully.

Materials and Methods

Why was the virtual electrode necessary? Please explain.

Was the virtual sensor in the above sentence the same meaning? “These “virtual sensor” time series were then...”???

Page 11

A line, “We here used a variant of this model where we increased the number of inhibitory units from three to 10 per layer”. Where the number three came from? Why was “three” inhibitory units not enough? The reason is provided such that the number of the principal cells would match to that of mini-column; however, alternatively, all could have been set to three. Why set to 10? Why not 3? Please provide a bit of background here.

A line, “To reduce the number of connection parameters...” why do they need to reduce the number of connection parameters? Please provide the reason.

A line, “The inversion of the neural mass model above uses the standard DCM approach...” Why “inversion” of the standard, DCM was used?

A line, “(e.g. inhibitory neurons do not genera a large dipole...)”. Please provide a reference or evidence for this.

Page 12

Description of fitting data from [16] is described. I might miss, but where is a description for fitting data from [15]?

A line, “we fitted all three stimuli conditions simultaneously by modelling difference sizes as ...” In the Method section, I would prefer to have more information on the “sizes”. Which sizes? “Parametric Empirical Bayes”. (PEB) is missing as you used this abbreviation below.

Page 13

A line, “...gamma frequency and amplitude”. Should this read, gamma “peak” frequency and amplitude?

Page 16 Reference, [36]-[41] is missing in the main text.

Page 17 Figure 1.

“All recurrent connection are inhibitory”. Please tell us why so?

“The same microcircuit was implemented both as a neural mass [8] and a compartmental model [5]”. An additional drawing or schematic would be helpful to picture what is microcircuit and what is neural mass (macroscale?) circuit.

Page 18 Figure 2.

What does the width of the horizontal arrows refer to??

In the figure, “IN” is drawn. In the caption, “Inhibitory Interneurons (II)”, instead. Please match to either IN or II.

Page 19 Figure 3.

A legend for (A) may be helpful.

“These quantify variability across subjects ...” Please be specific to what these error range (shade) corresponds to variability.

Please use a different colour on the shades, and provide which coloured shade correspond to which line.

Both colours are drawn in magenta, but I think using different colours may be easier to see the differences and overlap easier.

(B) 6 lines are drawn on the same panel, it may be good to show there are significant overlap across

conditions, yet, 6 lines in one panel may be too hard to see which one is which.

The numbers in the 4 panels in (B) look too small.

“Three stimulus sizes...” Please be specific which sizes?

(B), please explain which panel shows what kind of data is slotted.

Page 20 Figure 4.

Black/white scatter plots may be more straightforward. Please align the range of the Y-axes.

A connection (a11) is supplied; however, it is hard to understand what these labels correspond to what connection intuitively.

A vii) label is missing.

Page 20 Figure 4.

Along the x-axis, possibly a tick for 45Hz may help.

Dots and lines are plotted in magenta (to me). In the caption; however, it says fitted line shown in red. Please match. I am not sure if this simple scatter plot needs to be plotted in colour.

Page 21 Figure 5.

“line shown in red”. They look blue or purple for me. Please check.

Page 22 Figure 6.

“Evidence was very strong...” Which connection was very strong? Or the overall assessment, the results shown in this figure, all of them was strong? Please be specific.

Colours of arrows do not match. In (a), the thick arrow at a22 is in brown, but not orange as described.

Page 21 Table 1.

Please add a note why the unit for the microscale is μS , and that for macroscale is a.u.

Page 33 Figure S1.

Is this data from the previous study but not of your replication? I had also taken the data from [15] and plotted; however, $N = 16$ in their supplied results but the figure above contain only 14 data points. Am I wrong? If rejected a subject's data, please provide a reason and report so. Also, the correlation between V1 size and gamma peak frequency had been reported in [15] already. What was the intention to report the same results here, but now with different statistical results???

Reviewer #2 (Remarks to the Author):

Pinotsis and Miller present a simplified neural mass model for estimating the laminar dynamics that contribute to non-invasively recorded time frequency data. The current study presents evidence for deep cortical layers contributing to interindividual variability in visual gamma.

From a biological plausibility approach, I am interested in why the decision was made to use the model itself to explain V1 and visual cortex function, over the more complex but also more accurate to visual cortex CMC offered in DCM for example, with the granular layer. Do you have a reference to previous working demonstrating the superiority of this model at the non-invasive macroscale, such as by using BMS and the CMC model? Could this work be done if not?

The result and discussion are interesting – however there is a very heavy reliance on the results that

don't survive correction for multiple comparisons to substantiate the claims made, and the graphs in figure 4/5 show the data are not compelling – nor do they indicate that a trend is emerging. There is almost no interindividual variability in parameter modulation. For example figure 5: often if you take 3 participants out that would surely be an essentially straight line.

Unless I am mistaken and further description is needed, the suggestion that the reason the results don't survive correction is because the model was fit to non-invasive data is not compelling either to permit exploratory interpretation of the correlations. Instead it suggests the model may not have scaled up effectively to non-invasive data. There are examples in the literature of models that generate individual parameter estimates correlated onto visual gamma data features that produce biologically plausible results and survive multiple comparisons. For example (1).

The findings from the PEB analysis largely support the entire title/abstract/discussion and so are oddly placed as a small justification in the discussion for interpreting the correlations.

The most convincing and encouraging correlation is the relationship between parameter a_{41} and peak frequency in figure 4b – which given it's a PING mechanism predicted result makes sense (2). However, it is somewhat of concern that this or a similar superficial correlation does not emerge from the other dataset in Figure S2 to demonstrate that the model replicates commonly found superficial predictors of gamma, nor does any such relationship emerge from the PEB. To either discuss why this isn't a limitation, or acknowledge that it is, seems necessary.

Minor:

- Figure 1B – should include gamma on the x axis– given the paper is focused on gamma and to demonstrate whether the initial simulation included and fitted to an evident gamma bump.
- Least squares fitted lines aren't red in figure 4, 5, S2, or S3– fix the captions
- “To sum up” should only be used once per section else it's confusing to follow (see section: Computational models of brain dynamics at the micro and macro scales predict similar laminar data)
- Typo – last paragraph of methods – “without reinventing the model” do you mean reinventing?
- When referring to gamma peak in figure 4B, S2 A, add frequency to the title.
- The results/methods were difficult to follow – perhaps if the journal would allow, a paper that is so dependent on the methods would be better presented intro > methods > results> discussion
- When referring the reader to PEB – you cite 3 papers – the only one that is useful in this section is [10] “Intersubject variability and induced gamma in the visual cortex: DCM with empirical Bayes and neural fields”.

References

1. Shaw AD, Moran RJ, Muthukumaraswamy SD, Brealy J, Linden DE, Friston KJ, et al. Neurophysiologically-informed markers of individual variability and pharmacological manipulation of human cortical gamma. *Neuroimage*. 2017;161:19-31.
2. Tiesinga P, Sejnowski TJ. Cortical enlightenment: are attentional gamma oscillations driven by ING or PING? *Neuron*. 2009;63(6):727-32.

Reviewer #3 (Remarks to the Author):

Summary:

Interesting idea of combining 2 computational models of different spatial scales to explore visually induced gamma oscillations. One multi-compartment (micro-scale) model, including states such as receptor density. One (macroscale) neural mass model adapted from DCM. The crucial finding is that individual differences in oscillatory responses to visual stimuli are, in the context of this DCM model, best explained by inhibition in layer 5.

Overall paper is well written and does a good job of explaining a complex modelling approach in a fairly concise way. It definitely would be of interest to the neuro-modelling and broader electrophysiology readers. Minor comments, by section, are below.

I think the analysis and results are interesting and would recommend publication, following improvements to the manuscript. My major concerns are that there is a lot to explain and digest here, in terms of both methods and results, and this makes the clarity with which the script is written even more important than usual. Clarifying analysis steps; what was done in what order would help. Perhaps a figure summarising? – Not like figure 2, but giving an overview of the whole pipeline.

Intro:

I think the introduction does a good job of setting the rationale for the methods/analysis, although it can be quite hard to follow in places. Particularly the introduction of statistical decision theory suddenly switches from detailed to a very high-level description – e.g.

“In general, statistical decision theory prescribes the optimal way of using quantitative tools to make statistical decisions. It quantifies performance of these tools and what an optimal decision is. We found that in the context of statistical decision theory, compartmental and neural mass models can ...”

Given most readers will be numerically literate to some extent, can you instead give a description of SDT that gives the reader an intuition of what/how SDT works as a device for reconciling these models?

Modelling points / questions:

Both models parameterisation limited to the connectivity between layers.

If I've understood the steps correctly:

- You simulated power spectra from detailed micro-scale model and then fit the macro-scale model to these spectra using DCM SSR.
- Next, using the posterior parameters on the macro-model as priors, you fit 2 MEG datasets.
- The micro-scale model is never fit to data? Or the 'mapping' you generate between micro and macro isn't used to make some prediction about the parameters of the microscale model having fit the macro models to MEG data?

If this is correct, then the DCM of visually induced responses here is comparable to ref [18] Shaw et

al., but using a less V1-specific model. The correlations between parameters and gamma peaks (figure 4B) are very similar as those in [18] (<https://www.ncbi.nlm.nih.gov/pmc/articles/PMC5692925/figure/fig6/>) so some discussion of this – and the differences in key determinants of gamma between the studies would seem appropriate.

I think if you could really spell out each of the steps – like a timeline - of the analysis at the beginning of the results it would have a substantial impact on the clarity of the paper and help it to flow.

Results section ‘Computational models of brain dynamics at the micro and macro scales predict similar laminar data’:

Perhaps make clear at the beginning that this sub section is about creating a mapping between the micro- and macro-scale models, but not fitting MEG data / explaining individual diffs at this point. It’s not immediately obvious.

The penultimate and ultimate paragraphs both start ‘To sum up...’

Results section ‘Variability of visually induced gamma oscillations from different datasets reveals differences in the level of inhibition across different people’:

In this section the posteriors from the fitting above were used as priors to fit the macro-model to two separate MEG experiment datasets.

It is unclear from figure 3b whether the model has actually produced different fits for each of the 3 conditions – because there are so many lines on a small plot – perhaps it has, but the figure doesn’t make this clear; could it be configured or replotted to show this?

The DCM model here looks like an adaptation of the CMC, but where the L4 spiny stellates have been replaced with a second, deep interneuron population – presumably to align it with the micro-scale model. Is this a fair adjustment though, given the importance of L4 stellates in visual cortex, and their role of the major input point to the column? Without any context on why this was done, it seems like you’ve simply removed a crucial part of the V1 cytoarchitecture.

The correlations with V1 size are interesting but come out of nowhere; something should be added to the intro about this being a feature of your analysis!

Discussion:

This section does a good job of putting the results in the context of major hypotheses about gamma – however I think you should add references to the literature on the laminar generators of gamma & beta – e.g. the Traub/ Whittington /Koppel /Moghaddam type papers.

Methods & Materials:

The level of detail you describe the micro model in is much less than that of the macro/DCM model. The text suggests that the micro model is driven with a Gaussian input – in what domain – time? If

so, was the micro model integrated numerically? How did you get a spectral response? Was it the same method as the DCM?

DCM SSR uses a parameterised noise function (`spm_csd_mtf_gu`), which includes spectral shaping (i.e. $1/f$ -like) and neuronal fluctuations in frequency space. Was the micro-model subjected to this also? If not, are the spectral outputs of each really comparable?

A lot of effort is spent recapitulating parts of the DCM framework / equations, which emphasises how little information is provided about the microscale model methods. This balance needs addressing.

Equation 2 doesn't seem necessary, just cite one of the DCM papers. Or move to supplementary.

Equation 4 also seems supplementary worthy. It's doesn't add much and is a distraction.

Figures:

On your correlation scatter plots – both in the main text and supplementary figures – could you replace the y-label with an actual description of the parameter for readability? i.e. change 'alpha23' to 'L5 DP -> L5 II' or similar.

Differences in visually induced MEG oscillations reflect differences in deep cortical layer activity

Response to Reviewers' comments

We are grateful to the reviewers for their kind words, very helpful comments and careful reading of the paper. These comments have substantially improved the manuscript. We have taken all the reviewers' comments into consideration and have modified the text accordingly.

Please see below for our point-by-point responses to the reviewer comments.

(Reviewer comments in **bold**, responses in *italic* and changes to the m.s. denoted by track changes):

Reviewer #1:

Under the scope of individual differences in clinical heterogeneity, this article proposes the importance of advances in the neurocomputational model. Under the significant necessity of understanding of individual differences, this article aimed to deepen the knowledge of the variations on cortical gamma-band oscillation during visual perception. In previous works, it has been known that individual variations in the gamma oscillation were due to regional GABAergic concentration and the level of inhibitory inputs in the sources. It has been found that cortical inhibitory activity may contribute to the gamma oscillation. However, it is still not clear as to whether different layers (superficial and deep layers) at the macro- and micro-scale may play differential roles on the individual differences in the gamma oscillation. In this article, the authors modelled layered architecture-based estimation method based on non-invasively recorded (MEG) signals and fitted their model to the superficial and deep layers. Based on the observations, authors conclude that deep layer, but not superficial layer, of "both" macro and micro scale models, were the sources of the individual differences. Besides, it is claimed that combined modelling of both macro- and micro-scale model is necessary that was not sufficient with an isolated model.

The intent of the paper is quite essential to the applied neuroscience field. It is also worth giving credit the message that deepening the analytical model by incorporating macro- and micro-scale at different cortical layers would advance the field of research.

Our reply:

Thank you for the nice summary of our work above. Also, for the helpful remarks and suggestions below.

The method section seemed to be well-written and relatively easy to understand. In the meanwhile, the introduction and discussion were crude, and it was hard to read and follow the intent, lacking detailed explanation as to why the authors interpret the results. The set-up of the neural mass (macroscale) and microscale; these terms were used intermixed; it was a bit confusing. What was exactly how “macro” it was (see below). There are many places I have questions on, which can be described in detail, without skipping only to the main points. Also, the figures and the figure captions may be improved for the sake of ease of reading.

Our reply:

Thank you. We hope that our point to point responses and revisions to our ms detailed below have address these concerns.

They have re-analyzed two independent existing MEG data, while participants viewed visual stimuli. Induced power spectra data of 0.5–1.5 sec after stimulus onset were analyzed at the “virtual” sensor. But it was not clear why the virtual sensor conversion was necessary. Given the results that the spectra pattern observed between the macro- and micro-scale were the “similar”.

Our reply:

Sorry; we used virtual sensor data because model parameter estimates obtained here pertain to neural activity. This is part of standard DCM inference process where lead field (and observation model parameters in general) are estimated separately from neural parameters. This ensures that any effects found are due to changes in neural activity not confounds like volume conduction. We added on p. 11:

“In the second dataset, we used MEG beamformed data from [15]. Using virtual sensor or beamformed data allowed us to make inferences pertaining to neural activity (not the observation model).”

Yet it was not clear to me what it means in terms of cortical activity with respect to the “individual differences”, which was meant to be the purpose of the article.

Our reply:

Thank you. It simply means that we obtained virtual sensor data from single subjects.

Authors also mention about the relation to cortical volume in the V1 area as well as the E-I ratio that has been already proved by previous research. I understand varieties of the aspect of neural

markers are interrelated; however, I had kept distracted what was the main finding and message in this article. In addition, in the discussion, the authors proposes the future research will investigate E-I (excitatory-inhibitory) ratio. Yet given the current form of flows of their evidence and logic, they describe too far to mention the E-I ratio; such speculation need to be toned down as no direct evidence had been analyzed here on E-I. The results seem to support the authors claim; however, the way it is written was too simple. The authors may have some ideas and logic as to why the current results are so related to the individual differences in the E-I ratio observed in GABA, the quantity of evidence is lacking to support the speculation.

Our reply:

Thank you. In several DCM studies, parameter estimates pertaining to connections between excitatory and inhibitory populations of the sort obtained here have been interpreted as E-I balance proxies, e.g. [12], [40],[48-50]. We have now added on p. 10:

“Based on our results, we predict that E-I balance disruption will be more prominent in deep cortical layers. In future work, we will test this hypothesis More generally, obtaining connection parameter estimates informed by laminar dynamics allows one to consider several applications of DCM models where neural dynamics show interesting modulations depending on E-I balance, including circadian dynamics, working memory, neurological disorders etc, see e.g. [12], [40], and [48-50].”

Finally, the authors first propose an intent of developing a novel computational model to investigate the heterogeneity in clinical research populations; there are little connection I could find in these analyses and results.

Our reply:

This refers to the use of Parametric Empirical Bayes (PEB) in the last part of the Results section. This is a hierarchical Bayesian approach that has been used to study heterogeneity in large cohorts of healthy people and patients. We have now added further explanations and references on p. 8:

“ Specifically, we used a recent approach known as parametric empirical Bayes (PEB)(see Methods and Supplementary Methods for a summary of the theory behind this [20], [21]). This has been used to understand individual differences in effective connectivity in large cohorts of healthy people and patients [10],[20],[21] and [51].”

The most troublesome (for me to understand) was the 2x2 design of the depth of layers (superficial and deep) and scales (macro and micro). Notably, later in the text, the macro is expressed as lateral connection (should I interpret it as horizontal?). The term for macroscale, I would have expected more remotely interconnected network-level connection, whereas the micro is more regional and laminar (within a column). I think this could have been stated clearly from the beginning. Thus, the analysis was in the end, focusing only on the two columns in the visual cortex, which is very

regional to me. Authors could have stated that the validation was solely focusing only on the visual cortex, and this result may not necessarily generalize to the other regions or a network-level, etc.

Our reply:

Thank you. Yes, the terms horizontal and lateral connections are often used interchangeably in the literature. You are also right that whether our results hold in other areas is an open question. We have now added on p. 2:

“Here, we also extend the work of [8] from single subject animal data to multi-subject non-invasive human data. We analyze brain activity measured with MEG and consider between subject differences in visually induced gamma oscillations.

And then on p.9:

“Thus, our results suggest that differences in gamma oscillations between subjects originate from deep cortical layers. In future work, we will ask whether this holds in other areas beyond V1.”

To summarize my impression, all authors could re-write and reconsider the flows in the main introduction in a lay term as possible with enough background logic and explanations. Some changes in the figures may help readers to understand better.

Our reply:

Sure. Following your remarks here and below as well as other reviewers' remarks we have now rewritten the Introduction as follows (pp2-3):

“

We combine a model of neural compartments, describing dendrites and somata introduced in [5], with a biophysical neural mass model that predicts non-invasive brain data [6]. The model of [5] had been used to explain MEG oscillations in the alpha/beta [42] and gamma bands [14]. We used statistical decision theory [7] to prove that these two models can be combined to infer neural dynamics in different cortical layers (laminar dynamics) using non-invasive MEG data. In general, statistical decision theory (SDT) prescribes the optimal way of using quantitative tools to make statistical decisions in the face of uncertainty in the data [7]. This is often formulated in terms of decision rules. SDT has found applications in reinforcement learning [44] among other fields. Taking a statistical decision amounts to evaluating costs or losses based on same sample information combined with some other, e.g. prior or complementary, information. Here, we used SDT to reformulate compartmental and neural mass models as decision rules (besides other examples that mathematicians have considered as tools so far). Then, estimating neurobiological parameters of both compartmental and neural mass models is the same as making an optimal decision at different scales. After realizing this, we used

insights from SDT to estimate biophysically accurate parameter sets that describe neural dynamics at both the macro and micro scales.

In [8] and [9], we used a similar combination of computational models to analyze invasive animal data. However, we did not provide a mathematical proof of why such a combination can be considered and focused on invasive electrophysiology. Here we give a proof that establishes the mathematical basis of our approach. This also reveals limitations and suggests generalizations of our approach. Many alternative models can reproduce the same data (mean fields, neural masses, neural fields etc.). The proof reveals which of them can be thought of as equivalent (the ones where parameters can be thought of as statistical decision rule estimates). It also suggests a similar approach for multimodal datasets (where models correspond to e.g. fMRI and EEG) [41].

Here, we also extend the work of [8] from single subject animal data to multi-subject non-invasive human data. We analyze brain activity measured with MEG and consider between subject differences in visually induced gamma oscillations. Our aim was to quantify the neurobiological mechanisms that underlie variability in human MEG data. The extension from animal to human data, together with the earlier work of [8], suggest a two step approach for understanding dynamics and neurobiology at the microscale using non-invasive electrophysiology. Step one: Construct a mean field model that includes the same neuronal populations as a validated compartmental model that captures biophysical properties of single neurons (e.g., the geometry of the dendritic tree, kinetics and densities of ion channels, inputs from subcortical areas etc.) Fine tune its parameters to give similar predictions as the compartmental model. Test this with intracranial single subject recordings from non-human primates and rats. That was done in [8] and [9]. Step two: Test the same mean field model with human data. This is presented here.

Below, we illustrate our approach using two independent human MEG datasets analyzed earlier in [6] and [10]. These datasets contain visually induced oscillations recorded during perception tasks. In earlier work, we had found that differences in the above oscillations between different people were due to differences in the level of inhibition in the cortical source. To validate our new approach, we asked whether we could confirm those earlier results (obtained using different computational models). We also asked whether our new model of laminar dynamics could identify the cortical layer where these differences were more pronounced. To address these questions we computed correlations between model parameters describing laminar dynamics (connections) and V1 size that is known to predict gamma peak frequency [15]. Interestingly, both datasets led to the same result. Differences in visually induced oscillations between different people originated from differences in the level of inhibition in deep cortical layers. This provides new insights into the computations performed by different cortical layers.”

Below I note “subsets”, not all, of questions what I had to stop and think, read again back and forth, trying to understand what authors wanted to claim. There are too many sentences where I had to stop reading. If I need to evaluate this article again, I hope the future version may be easy to

read flawlessly so that a reviewer can fully engage in the judgment of the contents of the results better.

Page2, a paragraph starting with “Here, we also extend the work of [8]...”.

The description seems too simple, like a reading a list of brainstormed ideas. Authors could elaborate more on details.

Our reply:

Thank you. We have now rewritten parts of this paragraph, please see our response above.

Reviewer #1:

Page2, a paragraph starting with “Below, we demonstrate our approach using”.

A line, “we demonstrate our approach using two ...” It is awkward for me. Please rephrase.

A line, “we had found that differences in these oscillations...” Please be specific to what the “these” refers to.

A line, “To validate our new approach, we asked whether...” is stating the same meaning as the very first line of this paragraph. It is overlapping, so it may not be necessary.

A line, “... the cortical layer where these differences were more pronounced.” Please specify where was more pronounced?

Our reply:

Thank you. Following your suggestions, we have now rewritten this paragraph, please see our response above.

Reviewer #1:

Page 3 Results section,

The last sentence of the first paragraph, a sentence “...individual differences in single subject neurobiology...” What does it mean the “individual differences in a single subject”?

In the footer, 1, “Similar” also has a precise, mathematically rigorous, meaning. Where is the “1” on this page? I found the same sentence was described in the Methods.

Our reply:

Thank you. “1” is next to similar in the first paragraph of the section “Computational models of brain dynamics...” We also revisit some of this notions in Supplementary materials and methods in more detail.

We have also edited p.3 following your remarks above:

“Then we used hierarchical Bayesian modelling to explain individual differences in neurobiology and anatomy that underlie gamma oscillation variability.”

Reviewer #1:

Page 4,

A line, “Second, we simulated power spectra from the microscale model.” Please explain why you performed this. Please provide more explanation of why it needs to be done.

A line, “Predictions from both models are very similar”. Please provide statistical, quantitative proof.

A line, “The magenta and green curves overlap for frequencies above ... This difference in superficial layers is due to...” Why talked on an overlap and then all of the sudden talks about a difference?

The last line of the paragraph, “This difference in superficial layers is due to the fact that ...” where can I find the fact? Please provide a point of reference. I would appreciate a more detailed explanation here.

A line, “... the same frequency peaks and similar power distribution across frequencies in superficial and deep layers.” It was not clear to me why this awkward way to analysis was needed by adapting the microscale model and a macroscale model that is describing the same microcircuit... please provide logic for why adapting one to the other was necessary.

A title, “Variability of visually induced gamma oscillations from difference datasets...” Please be specific on “datasets”. May it be better to replace “datasets” with “visual stimuli on different observers”?

Our reply:

Sure, no problem. We have now incorporated your suggestions in the updated ms (p.4):

“Besides describing the same microcircuit, the macro- and microscale models predict similar laminar dynamics. The mathematical proof of the equivalence of their predictions is based on

statistical decision theory and is included in the Supplementary Materials and Methods. To make the two models predict similar dynamics, we used an analysis pipeline that combines existing and validated methodologies (Figure 2) introduced in [8]. In brief, this pipeline includes the following 4 steps: 1. Simulate data from the compartmental model. 2. Fit the mass model to these data. 3. Use the parameter estimates obtained as priors for fitting M/EEG data. 4. Obtain hidden parameters that describe laminar dynamics. We discuss them below. Here, this pipeline is also justified using theoretical arguments from statistical decision theory. This theory required that the micro- scale model be adapted to have the same number of parameters as the macroscale model. We changed the compartmental model of [5], by reducing the number of its connection parameters (considering all synaptic weights equal) and called the resulting model the symmetric compartmental model see also [8], [9]¹. The remaining parameters are the weights of the connections between neural populations occupying different layers of the microcircuit shown in Figure 1A. These are the same parameters that the macroscale model has. We then simulated power spectra from the microscale (symmetric compartmental) model (green data in Figure 1B) and fitted the macroscale model to them using Dynamic Causal Modelling (DCM) for steady state responses [9], [11], [12]. This ensured the neural mass model has construct validity in relation to the compartmental model. DCM approach has been used to infer changes in synaptic plasticity in large cortical networks [11] in healthy and clinical populations [13] among other applications. It exploits the stationarity of variance of neuronal firing over the course of the task to efficiently fit neural mass models to MEG data. After fitting, we obtained the parameter values of the macroscale model that fitted best the predictions of the microscale model. The predictions from the macroscale model for these parameter values are shown in Figure 1B in magenta. Predictions from both models are very similar. The correlation between these predictions is $r = 0.5$, $p < 10^{-3}$ for power spectra of superficial pyramidal cells and $r = 0.75$, $p < 10^{-4}$ for power spectra of deep pyramidal cells. The magenta and green curves overlap for frequencies above 3 Hz for deep layers (Figure 1B bottom) and 8Hz for superficial layers (Figure 1B top). This difference in superficial layers is due to the fact that the macroscale model receives explicit white and pink noise input that propagates to all layers, while the microscopic models received Gaussian spike input targeting superficial layers, see [9] and [42] for more details. Exogenous input to the compartmental model groups of 10 bursts (each consisting of 2 spikes separated by a 10 ms interval) with 100 ms intervals between the burst groups. Despite these differences, we show below that SDT allows us to estimate neural mass model parameters for which the neural mass predicts the same neural dynamics as the compartmental model. This is done by fitting the neural mass to simulated data from the compartmental model.

To sum up, we adapted the microscale model of [5] and fine-tuned the parameters of a macroscale model describing the same microcircuit so that the two models make the predictions of the same frequency peaks and similar power distribution across frequencies in superficial and deep layers. The two models are thus functionally equivalent. The parameter values of the two models for which this happens are included in Table 1. Parameters for the microscale model are

¹ In (Pinotsis et al., 2017a), we found that evoked responses from the original model of [5] and its symmetric variant were highly correlated, $r=0.9343$, $p < 0.001$, see the Supplementary Figure S1A.

shown in the left column while the parameters of the macroscale model in the right. These include the connection parameters between the populations shown in Figure 1: inhibitory interneurons in superficial layers, SI; pyramidal cells in superficial layers, SP; inhibitory interneurons in deep layers, DI; pyramidal cells in deep layers, DP. Each parameter corresponds to one arrow. There are in total ten connections in both the microscale and macroscale models.”

Reviewer #1:

Page 8, In Discussion,

A line, “we found that individual differences in gamma oscillations reflected the level of inhibition in the cortical source.” Does it mean, this article simply replicated the previous study, [6]? I do not see much differences in the sentences between the very first sentence of the same paragraph (“We found that individual differences in visually induced ...”)and this line.

In the second paragraph, “subjects’ responses originate”. Does it mean subjects’ neural responses originate? I hardly believe subjects’ physical response (like pressing buttons) originate from the visual cortex. Please be specific.

Our reply:

Thank you. Yes, part of our results confirmed our earlier work. We have also edited our m, like you suggested:

“we were able to locate the cortical layer that might be the origin of differences between oscillatory responses of different subjects.”

Reviewer #1:

Page 9.

In a paragraph starting “Third: ...”

A line “(1) difference processing of the same sensory input across people who differ in their predictions that ...” In this paragraph, describing a link to predictive coding theories. I was not entirely convinced on this line. Please provide more explanations.

A line, “Here, we found that deep cortical inhibitions are also related to gamma oscillation variability ...” Again, I am not convinced how are they related. Please provide more explanations.

A line, “We also revealed differences in brain structure and function between different people.” I was not sure if any structure (grey matter volume) was related to the function in this article?

Our reply:

Thank you. We now write on this page:

“PC suggests that deep layer activity corresponds to different hypotheses about sensory input that different people have [11], [31]. Deep layers are thought to represent expectations of sensory input. Different expectations can in turn, be due to different prior experiences.”

Also:

“Here, we found that deep cortical inhibition is also related to gamma oscillation variability observed in human subjects using correlations and a greedy search. Specifically, we found that deep cortical inhibition parameters correlated with V1 size. This in turn, was shown to correlate with gamma oscillation variability [15].”

And:

“We also revealed differences in brain structure (V1 size) and inhibitory function between different people [10].”

Reviewer #1:

Page10

A line, “At the same time, it is expressed in the microscale as e.g. differences in the E-I balance. Thus, we need multiscale approaches to study...” I did not understand why we need a multiscale approach at all. Please describe. Same for the line, “Using isolated models is not enough”. Please kindly refer to where I can find the evidence that the isolated model was not enough. If not enough, how much does it lack compared to the multiscale approach, where I can see the quantitative results?

A line, “Spatial information at the macroscale is accessible via non-invasive human brain imaging. Spatial information at the microscale and the structure and function of local cortical circuits is accessible in vitro or through invasive recordings in animals.” I understand that proposal of a combination may help translate animal data at the microscale to macro-level human or network-level data. However, to be simple, collection of both data from the same individual in vivo may provide the best result rather than combining animal data to a human directly. Please provide more logic here to convince readers. Just stating “combination should be considered” without proof may not be enough to persuade readers fully.

Our reply:

Thank you. We simply meant that it is not possible to gather simultaneous non-invasive and invasive (macro and microscale) data for most human subjects with the exception of perhaps epileptic patients – which is a special case with its own limitations. If the reviewer has another idea, we would be happy to consider this. This is the first approach that used a combination of models at different spatial scales to establish the construct validity of a mean field (neural mass) model in relation to a compartmental model.. The reviewer might consult any of the relevant references in the literature. Following your suggestions, we now write on this page:

“Heterogeneity of neurological diseases and disorders is studied using non invasive MEG data that sample brain responses at the macroscale. At the same time, it is expressed in the microscale (cortical circuit level) as e.g. differences in the E-I balance. Thus, we need *multiscale* approaches to study this heterogeneity.”

Reviewer #1:

Materials and Methods

Why was the virtual electrode necessary? Please explain.

Was the virtual sensor in the above sentence the same meaning? “These “virtual sensor” time series were then...”???

Our reply:

Thank you. Yes, they are the same. We have now replaced electrode with sensor throughout the ms to avoid any confusion. Regarding the meaning and use of virtual sensor data please see our response to your earlier point above.

Reviewer #1:

Page 11

A line, “We here used a variant of this model where we increased the number of inhibitory units from three to 10 per layer”. Where the number three came from? Why was “three” inhibitory units not enough? The reason is provided such that the number of the principal cells would match to that of mini-column; however, alternatively, all could have been set to three. Why set to 10? Why not 3? Please provide a bit of background here.

A line, “To reduce the number of connection parameters...” why do they need to reduce the

number of connection parameters? Please provide the reason.

A line, “The inversion of the neural mass model above uses the standard DCM approach...” Why “inversion” of the standard, DCM was used?

A line, “(e.g. inhibitory neurons do not genera a large dipole...)”. Please provide a reference or evidence for this.

Our reply:

Thank you. Following your suggestions, we now write on this page:

“We here used a variant of this model where we increased the number of inhibitory units from three to 10 per layer, so that their number was equal to the number of the principal cells within each mini-column. The original model comprised 10 PNs in layers 2/3, 10 PNs in layer 5, and 3 INs in both layers. The synaptic architecture followed general tenets of cortical micro-circuitry where FF connections target the granular layer and FB connections target agranular layers, see e.g. (Pinotsis et al., 2013) for a further discussion. Modelling of single neuron morphology and physiology followed (Bush and Sejnowski, 1993), using the same parameters as in (Jones et al., 2007). Details can be found in (Pinotsis et al., 2017). To ensure that relative differences in interneuron densities were accommodated, we multiplied the maximum conductance values of the corresponding connections by a factor of 0.3. To reduce the number of connection parameters (synaptic strengths) to be equal to those of the macroscopic model, we assumed that the connection weights between different mini-columns comprising the microcircuit shown in Figure 1A were the same. This assumed symmetry constraints on horizontal connectivity (within each cortical layer) of the sort assumed in mean field models that describe aggregate activity over hundreds of neurons. We then simulated data from the symmetric compartmental model and made them amenable to a further DCM analysis, see *simData.mat* in the above Github repository. The model was integrated using the implicit functionality of NEURON. We then used a simple Welch method for obtaining spectral density estimates. This was also used in DCM. [...]

These contributions are based on differences in anatomical properties and the lead field configuration of each population (e.g. inhibitory neurons do not generate a large dipole[52])...”

Reviewer #1:

Page 12

Description of fitting data from [16] is described. I might miss, but where is a description for fitting

data from [15]?

A line, “we fitted all three stimuli conditions simultaneously by modelling difference sizes as ...” In the Method section, I would prefer to have more information on the “sizes”. Which sizes? “Parametric Empirical Bayes”. (PEB) is missing as you used this abbreviation below.

Page 13

A line, “...gamma frequency and amplitude”. Should this read, gamma “peak” frequency and amplitude?

Our reply:

Thank you. The stimuli were described on p.5:

“We first fitted the macroscale model to the MEG data from [16]. These included visually induced oscillations while 12 subjects viewed grating stimuli of three different sizes— 2° , 4° & 8° . Thus the dataset from [16] included oscillations induced by three different stimulus sizes.”

Following your suggestions, we now write on p. 14:

“Using DCM, we fitted the neural mass model to power spectra between 30-80Hz from different subjects reported in [15], [16]. This frequency range contains gamma frequencies captured with MEG[15], [16]. This analysis was similar to our earlier work described in [6], [10] where a detailed description of the DCM procedure is given. To fit data from [16], we fitted all three stimuli conditions (different stimulus size described above and in [16]) simultaneously by modeling different sizes as condition-specific effects (the B matrix in DCM), see [10]. We next consider how this likelihood model is placed within a hierarchical model of responses from multiple subjects.

Between subject differences. To model between-subject effects we used a hierarchical Bayesian inference approach known as Parametric Empirical Bayes (PEB), see [10] and the function `spm_dcm_peg_bmc.m` in the DCM toolbox. At the first level, we used the neural mass model described above and in Results. At the second level, we modelled individual differences in gamma peak frequency and amplitude...”

Reviewer #1:

Page 16 Reference, [36]-[41] is missing in the main text.

Page 17 Figure 1.

“All recurrent connection are inhibitory”. Please tell us why so?

“The same microcircuit was implemented both as a neural mass [8] and a compartmental model

[5]”. An additional drawing or schematic would be helpful to picture what is microcircuit and what is neural mass (macroscale?) circuit.

Our reply:

These references are included in the Supplementary Material. The circuit is the same and shown in Figure 1A. We have also edited the figure legend like you suggested:

“Arrows denote excitatory and inhibitory connections. All recurrent connections are inhibitory to preclude run-away excitation in the network. The same microcircuit was implemented both as a neural mass [8] and a compartmental model [5].”

Reviewer #1:

Page 18 Figure 2.

What does the width of the horizontal arrows refer to??

In the figure, “IN” is drawn. In the caption, “Inhibitory Interneurons (II)”, instead. Please match to either IN or II.

Our reply:

The width corresponds to the connection strength and delay effects. The legend reads as follows: “Here horizontal arrows of different widths in the left panel denote asymmetric connectivities and delays between mini-columns depicted as rectangles containing Superficial and Deep Pyramidal cells (SP and DP) and Inhibitory Interneurons (IN).”

Reviewer #1:

Page 19 Figure 3.

A legend for (A) may be helpful.

“These quantify variability across subjects ... ” Please be specific to what these error range (shade) corresponds to variability.

Please use a different colour on the shades, and provide which coloured shade correspond to which line.

Both colours are drawn in magenta, but I think using different colours may be easier to see the differences and overlap easier.

(B) 6 lines are drawn on the same panel, it may be good to show there are significant overlap across conditions, yet, 6 lines in one panel may be too hard to see which one is which.

The numbers in the 4 panels in (B) look too small.
 “Three stimulus sizes...”. Please be specific which sizes?
 (B), please explain which panel shows what kind of data is slotted.

Our reply:

Variability is across subjects. Following your suggestion, we have also enlarged the size of B to make the different lines and numbers more visible. Like we said in our response above, stimulus details and sizes are described in [16]. Different panels show data from four different anonymous subjects.

Reviewer #1:

Page 20 Figure 4.

Black/white scatter plots may be more straightforward. Please align the range of the Y-axes.
 A connection (a11) is supplied; however, it is hard to understand what these labels correspond to what connection intuitively.
 A vii) label is missing.

Our reply:

Sorry; we have now added the missing label and kept the colors as we refer to them in the ms to distinguish between different results obtained. Regarding your remark about the meaning of labels, you are right: we have now included an explicit description of each parameter in a new column as part of Table 1:

Table 1 Synaptic connectivity parameters for the microscale and macroscale models.

	Description	Max conductance (μ S) Microscale model	Intrinsic connectivity Macroscale model (a.u)
a_{44}	SP→SP	0.001	4.4
a_{14}	SP→SI	0.003	4.8
a_{34}	SP→DP	0.00025	23.3
a_{41}	SI→SP	0.015	3.8
a_{31}	SI→DP	0.0003	5.9
a_{11}	SI→SI	0.0006	4.2
a_{33}	DP→DP	0.005	2.2
a_{23}	DP→DI	0.0003	4.6

a_{32}	DI->DP	0.0075	6.9
a_{22}	DI->DI	0.0006	4.16

Reviewer #1:

Page 20 Figure 4.

Along the x-axis, possibly a tick for 45Hz may help.

Dots and lines are plotted in magenta (to me). In the caption; however, it says fitted line shown in red. Please match. I am not sure if this simple scatter plot needs to be plotted in colour.

Our reply:

Sorry; we have now corrected this.

Reviewer #1:

Page 21 Figure 5.

“line shown in red”. They look blue or purple for me. Please check.

Our reply:

Our apologies; we have now corrected this too.

Reviewer #1:

Page 22 Figure 6.

“Evidence was very strong...” Which connection was very strong? Or the overall assessment, the results shown in this figure, all of them was strong? Please be specific.

Colours of arrows do not match. In (a), the thick arrow at a22 is in brown, but not orange as described.

Our reply:

Sure; we have now updated the legend as follows:

“We scored alternative GLMs where predictors of variability in V1 included any combination of the connections (arrows) in Fig.1A. We found that for the data from [15] V1 size could

be best predicted by the recurrent connectivity of deep inhibitory interneurons, a_{22} (brown arrow). Evidence in favour of a GLM including a_{22} was very strong $p>0.95$. **(B)** Same as in (A) for data from [16]. V1 size variability reported in [16] could be best predicted by the inhibitory drive to deep pyramidal cells, a_{31} (brown arrow). Evidence for the corresponding GLM was weak $p>0.5$.”

Reviewer #1:

Page 21 Table 1.

Please add a note why the unit for the microscale is uS, and that for macroscale is a.u.

Reviewer #1:

Our reply: Sure. We have added to the title “Units follow standard conventions in the literature.” These are well known.

Page 33 Figure S1.

Is this data from the previous study but not of your replication? I had also taken the data from [15] and plotted; however, N =16 in their supplied results but the figure above contain only 14 data points. Am I wrong? If rejected a subject’s data, please provide a reason and report so. Also, the correlation between V1 size and gamma peak frequency had been reported in [15] already. What was the intention to report the same results here, but now with different statistical results???

Our reply:

Sorry; you are right. Two subjects from the original pool were discarded due to poor V1 size measurements. The results shown in Figure S1 are not different, they are the same as before (the figure has been redrawn following the suggestion of another reviewer). Note that the original paper [15] has a very similar figure –which might the origin of the confusion. In that figure, correlations are different. The reason for that is that that figure includes slightly different gamma peaks (average of peaks in hemispheres corresponding to the left and right visual fields) while we only considered peaks in one hemifield, following the virtual sensor analysis. Nevertheless all peaks give significant correlations. We have now added on p. 36:

“Strong positive correlation between the size of primary visual cortex in a certain subject and the corresponding gamma peak frequency ($R=0.727$, $p=0.003$) reported in [15]. Data from two subjects among the original pool were discarded due to poor quality of the corresponding V1 size measurement.”

We hope that the above changes and additions to the manuscript were what you had in mind.

Reviewer #2:

Pinotsis and Miller present a simplified neural mass model for estimating the laminar dynamics that contribute to non-invasively recorded time frequency data. The current study presents evidence for deep cortical layers contributing to interindividual variability in visual gamma.

From a biological plausibility approach, I am interested in why the decision was made to use the model itself to explain V1 and visual cortex function, over the more complex but also more accurate to visual cortex CMC offered in DCM for example, with the granular layer. Do you have a reference to previous working demonstrating the superiority of this model at the non-invasive macroscale, such as by using BMS and the CMC model? Could this work be done if not?

Our reply:

Thank you for the helpful remarks and suggestions here and below. Regarding your question, indeed the model used here is a variant of the CMC model. It is the neural mass analogue of a compartmental model developed by (Jones et al., 2007). Previous work by Stephanie Jones and colleagues has validated this model using MEG data: the model has successfully explained brain dynamics including evoked responses (Jones et al., 2007), rhythmogenesis and the alpha and beta oscillations (Jones et al., 2009) and also gamma oscillations (Jones et al., 2013). This model includes detailed single neuron morphology and physiology as well intricate description of current dipole sources that underlie MEG signals of the sort considered here. Our approach is based on analysing a compartmental and a neural mass model together. To date, there is no compartmental model based on the CMC neural mass. Thus, we could not had used the CMC model. Constructing and validating a compartmental CMC model would be a long-term project –possibly resulting in several distinct publications. Like the other reviewer also pointed out, the CMC and the (Jones et al., 2007) models are very similar. The main difference is a distinct spiny stellate cell population. This is important, however gamma oscillations that are considered here have been shown to be primarily dependent on parameters of other populations (superficial pyramidal cells and inhibitory interneurons) that are present in both models. We have added to the manuscript (p. 2):

“We combine a model of neural compartments, describing dendrites and somata introduced in [5], with a biophysical neural mass model that predicts non-invasive brain data [6]. The model of [5] had been used to explain MEG oscillations in the alpha/beta [42] and gamma bands [14]. We used statistical decision theory [7] to prove that the these two models can be combined to to infer neural dynamics in different cortical layers (laminar dynamics) using non-invasive MEG data.”

Reviewer #2:

The result and discussion are interesting – however there is a very heavy reliance on the results that don’t survive correction for multiple comparisons to substantiate the claims made, and the graphs in figure 4/5 show the data are not compelling – nor do they indicate that a trend is emerging. There is almost no interindividual variability in parameter modulation. For example figure 5: often if you take 3 participants out that would surely be an essentially straight line.

Unless I am mistaken and further description is needed, the suggestion that the reason the results don't survive correction is because the model was fit to non-invasive data is not compelling either to permit exploratory interpretation of the correlations. Instead it suggests the model may not have scaled up effectively to non-invasive data. There are examples in the literature of models that generate individual parameter estimates correlated onto visual gamma data features that produce biologically plausible results and survive multiple comparisons. For example (1).

Our reply:

The reviewer is right. Earlier work has found stronger correlations. We have now included references to that. Interestingly, in those analyses, prior estimates of connection parameters did not include information about differences in neural activity between different cortical layers, contrary to our current analyses. By constraining our mass model to satisfy laminar constraints dictated by the compartmental model, we have lost the flexibility that previous models had while explaining MEG data, that, in any case contain limited information about laminar dynamics. We now write on p.7:

“However, all but one correlation did not remain significant when corrected for multiple comparisons using a Bonferroni correction or accounting for other sources of variance using partial correlations. Previous work with a similar model had found correlations between parameters and gamma oscillation features that survived multiple comparisons [18]. This is interesting and could be explained by the fact that prior parameters in that earlier work did not include information about differences in neural activity between different cortical layers. This rendered that earlier model are more flexible while explaining MEG data. Constraining parameters to differentiate between superficial and deep layer activity like we did here poses an extra challenge given that information about laminar (depth) differences in neural activity might be limited in MEG data.”

The findings from the PEB analysis largely support the entire title/abstract/discussion and so are oddly placed as a small justification in the discussion for interpreting the correlations.

Our reply:

Sorry; we just wanted to motivate the PEB analysis by considering the most straightforward approach first, that is, computing correlations. We devoted the last part of the second section of Results that focuses on correlations (“Variability of visually induced gamma oscillations from different datasets reveals differences in the level of inhibition across different people”) to precisely this: to illustrate the conceptual link between correlations and PEB. This is to help new readers understand PEB. We show that PEB can address questions similar to classical statistics and it might offer an advantage. In this setting, a probabilistic model downweighs estimates from subjects that might be outliers. We also devoted the last section of the Results (“Individual differences in MEG oscillations reflect differences in V1 size and deep layer activity”) to a thorough discussion of the PEB results. Should the reviewer think that this should be changed, we are happy to do so.

The most convincing and encouraging correlation is the relationship between parameter a41 and

peak frequency in figure 4b – which given it’s a PING mechanism predicted result makes sense (2). However, it is somewhat of concern that this or a similar superficial correlation does not emerge from the other dataset in Figure S2 to demonstrate that the model replicates commonly found superficial predictors of gamma, nor does any such relationship emerge from the PEB. To either discuss why this isn’t a limitation, or acknowledge that it is, seems necessary.

Our reply:

Thank you. Indeed, this is an interesting correlation. Note that it also survives the correction for multiple comparisons. The reviewer is also right. These limitations should be acknowledged. We have now done this (p.7):

“ No changes were significantly correlated. Also, a_{41} did not correlate with peak frequency in this dataset.”

And referenced prior work on p.8:

“Gamma peak frequency was not predicted by connection changes, like a_{41} that is known to underlie the PING mechanism [43].”

Reviewer #2:

Minor:

- Figure 1B – should include gamma on the x axis– given the paper is focused on gamma and to demonstrate whether the initial simulation included and fitted to an evident gamma bump.

Our reply:

Sorry; we followed the original publications (Jones et al., 2007; 2009) and included only low gamma (30-50Hz). This is based on the model available on ModelDB database – so that other researchers can use our approach. We have now added to p.3:

“The model predicts the same peak frequencies for superficial and deep pyramidal cells and similar power distribution across frequencies (Figure 1B) as the compartmental model available in the ModelDB database², including alpha, beta and low gamma.”

Reviewer #2:

- Least squares fitted lines aren’t red in figure 4, 5, S2, or S3– fix the captions

2

https://senselab.med.yale.edu/modeldb/ShowModel.cshtml?model=136803&file=/JonesEtAl2009/mod_files/km.mod#tabs-2

Our reply:

Sorry; we have now corrected this.

Reviewer #2:

- **“To sum up” should only be used once per section else it’s confusing to follow (see section: Computational models of brain dynamics at the micro and macro scales predict similar laminar data)**

Our reply:

We have now corrected this. Thank you.

Reviewer #2:

- **Typo – last paragraph of methods – “without reinventing the model” do you mean reinverting?**

Our reply:

Sorry; this has now been corrected.

Reviewer #2:

- **When referring to gamma peak in figure 4B, S2 A, add frequency to the title.**

Our reply:

Sure; we have now added this.

Reviewer #2:

- **The results/methods were difficult to follow – perhaps if the journal would allow, a paper that is so dependent on the methods would be better presented intro > methods > results> discussion**

Our reply:

Sure; if the journal allows this, we would be happy to have the sections rearranged like you suggested.

Reviewer #2:

- **When referring the reader to PEB – you cite 3 papers – the only one that is useful in this section is [10] “Intersubject variability and induced gamma in the visual cortex: DCM with empirical Bayes and neural fields”.**

Our reply:

Sure; we have now removed the extra two papers like you suggested to emphasise affinity and direct the reader (and included them as general references below to acknowledge colleagues' work).

References

- 1. Shaw AD, Moran RJ, Muthukumaraswamy SD, Brealy J, Linden DE, Friston KJ, et al. Neurophysiologically-informed markers of individual variability and pharmacological manipulation of human cortical gamma. Neuroimage. 2017;161:19-31.**
- 2. Tiesinga P, Sejnowski TJ. Cortical enlightenment: are attentional gamma oscillations driven by ING or PING? Neuron. 2009;63(6):727-32.**

Our reply:

Thank you. We have now included both references above.

We hope that the above changes and additions to the manuscript were what you had in mind.

Reviewer #3:

Summary:

Interesting idea of combining 2 computational models of different spatial scales to explore visually induced gamma oscillations. One multi-compartment (micro-scale) model, including states such as receptor density. One (macroscale) neural mass model adapted from DCM. The crucial finding is that individual differences in oscillatory responses to visual stimuli are, in the context of this DCM model, best explained by inhibition in layer 5.

Overall paper is well written and does a good job of explaining a complex modelling approach in a fairly concise way. It definitely would be of interest to the neuro-modelling and broader electrophysiology readers. Minor comments, by section, are below.

Our reply:

Thank you for the kind words as well as helpful remarks and suggestions below.

Reviewer #3:

I think the analysis and results are interesting and would recommend publication, following improvements to the manuscript. My major concerns are that there is a lot to explain and digest here, in terms of both methods and results, and this makes the clarity with which the script is written even more important than usual. Clarifying analysis steps; what was done in what order would help. Perhaps a figure summarising? – Not like figure 2, but giving an overview of the whole pipeline.

Our reply:

Thank you. We agree, this would be very helpful to the reader. We have now added a new Figure as part of what was before Figure 2 (Figure 2A). These summarizes the analysis steps of our pipeline (please see also our response to your remarks on Modelling points below):

We also revised the legend of Figure 2 as follows:

Fig. 2. (A) Schematic of our analysis pipeline. This summarizes the steps of our approach: 1. Simulate data from the compartmental model. 2. Fit the mass model to these data. 3. Use the parameter estimates obtained as priors for fitting M/EEG data. 4. Obtain hidden parameters that describe laminar dynamics. **(B)** Construction of the neural mass model: We first establish a similarity between the model of [5] and its symmetric variant. Here horizontal arrows of different widths in the left panel denote asymmetric connectivities and delays between mini-columns depicted as rectangles containing Superficial and Deep Pyramidal cells (SP and DP) and Inhibitory Interneurons (II). In the right panel, a symmetrisation of the compartmental model reduces the number of connectivity parameters to be the same as those in a homologous neural mass model. **(C)** Construct validity of the mass model. To demonstrate this, we fitted the mass model to synthetic (laminar) data obtained from its compartmental homologue. This is justified by statistical decision theory. Red and green lines in the middle panel correspond to real data and model predictions. Solid and dashed lines to real and imaginary parts of cross spectra between deep and superficial pyramidal neurons.”

Finally, we added the following on p.4:

“To make the two models predict similar dynamics, we used an analysis pipeline that combines existing and validated methodologies (Figure 2) introduced in [8]. In brief, this pipeline includes the following 4 steps: 1. Simulate data from the compartmental model. 2. Fit the mass model to these data. 3. Use the parameter estimates obtained as priors for fitting M/EEG data. 4. Obtain hidden parameters that describe laminar dynamics. We discuss them below. Here, this pipeline is also justified using theoretical arguments from statistical decision theory.”

Reviewer #3:

Intro:

I think the introduction does a good job of setting the rationale for the methods/analysis, although it can be quite hard to follow in places. Particularly the introduction of statistical decision theory suddenly switches from detailed to a very high-level description – e.g.

“In general, statistical decision theory prescribes the optimal way of using quantitative tools to make statistical decisions. It quantifies performance of these tools and what an optimal decision is. We found that in the context of statistical decision theory, compartmental and neural mass models can ...”

Given most readers will be numerically literate to some extent, can you instead give a description of SDT that gives the reader an intuition of what/how SDT works as a device for reconciling these models?

Our reply:

Sure, thank you. We have now added on p. 2:

“ We used statistical decision theory [7] to prove that the these two models can be combined to to infer neural dynamics in different cortical layers (laminar dynamics) using non-invasive MEG data. In general, statistical decision theory (SDT) prescribes the optimal way of using quantitative tools to make statistical decisions in the face of uncertainty in the data [7]. This is often formulated in terms of decision rules. SDT has found applications in reinforcement learning [44] among other fields. Taking a statistical decision amounts to evaluating costs or losses based on same sample information combined with some other, e.g. prior or complementary, information. Here, we used SDT to reformulate compartmental and neural mass models as decision rules (besides other examples that mathematicians have considered as tools so far). Then, estimating neurobiological parameters of both compartmental and neural mass models is the same as making an optimal decision at different scales. After realizing this, we used insights from SDT to estimate biophysically accurate parameter sets that describe neural dynamics at both the macro and micro scales.”

Reviewer #3:

Modelling points / questions:

Both models parameterisation limited to the connectivity between layers.

If I've understood the steps correctly:

- You simulated power spectra from detailed micro-scale model and then fit the macro-scale model to these spectra using DCM SSR.**
- Next, using the posterior parameters on the macro-model as priors, you fit 2 MEG datasets.**
- The micro-scale model is never fit to data? Or the ‘mapping’ you generate between micro and macro isn’t used to make some prediction about the parameters of the microscale model having fit the macro models to MEG data?**

If this is correct, then the DCM of visually induced responses here is comparable to ref [18] Shaw et

al., but using a less V1-specific model. The correlations between parameters and gamma peaks (figure 4B) are very similar as those in [18]

(<https://www.ncbi.nlm.nih.gov/pmc/articles/PMC5692925/figure/fig6/>)

so some discussion of this – and the differences in key determinants of gamma between the studies would seem appropriate.

I think if you could really spell out each of the steps – like a timeline - of the analysis at the beginning of the results it would have a substantial impact on the clarity of the paper and help it to flow.

Our reply:

Thank you. Your summary of steps was absolutely right. Following your suggestion, we have now included these steps in Figure 2A (please see also our response to your comments above). Also your remark about the Shaw et al model is correct too ([18] in our reference list). That model includes a similar cortical circuitry and connections. We now elaborate on similarities and differences further. We have added the following on p.7:

“However, all but one correlation did not remain significant when corrected for multiple comparisons using a Bonferroni correction or accounting for other sources of variance using partial correlations. Previous work with a similar model had found correlations between parameters and gamma oscillation features that survived multiple comparisons [18]. This is interesting and could be explained by the fact that prior parameters in that earlier work did not include information about differences in neural activity between different cortical layers. This rendered that earlier model more flexible while explaining MEG data. Constraining parameters to differentiate between superficial and deep layer activity like we did here poses an extra challenge given that information about laminar (depth) differences in neural activity might be limited in MEG data. “

Reviewer #3:

Results section ‘Computational models of brain dynamics at the micro and macro scales predict similar laminar data‘:

Perhaps make clear at the beginning that this sub section is about creating a mapping between the micro- and macro-scale models, but not fitting MEG data / explaining individual diffs at this point. It’s not immediately obvious.

Our reply:

Sure; we have now added on p. 3:

“First we linked predictions of brain dynamics at different scales. This entailed a mapping between micro- and macro-scale models that we discuss below. Following [8], we here used a macroscale (neural mass) model whose parameters had been tuned ...”

Reviewer #3:

The penultimate and ultimate paragraphs both start ‘To sum up...’

Our reply:

Sorry; we have now corrected this.

Reviewer #3:

Results section ‘Variability of visually induced gamma oscillations from different datasets reveals differences in the level of inhibition across different people‘:

In this section the posteriors from the fitting above were used as priors to fit the macro-model to two separate MEG experiment datasets.

It is unclear from figure 3b whether the model has actually produced different fits for each of the 3 conditions – because there are so many lines on a small plot – perhaps it has, but the figure doesn’t make this clear; could it be configured or replotted to show this?

Our reply:

Sorry; Indeed, differences are tiny. However, the model fits the conditions separately. We have now enlarged the Figure 3B. We show the full 30-80Hz range of gamma oscillations—this constrained the range of y-values that can be shown (given the overall figure size).

Reviewer #3:

The DCM model here looks like an adaptation of the CMC, but where the L4 spiny stellates have been replaced with a second, deep interneuron population – presumably to align it with the micro-scale model. Is this a fair adjustment though, given the importance of L4 stellates in visual cortex, and their role of the major input point to the column? Without any context on why this was done, it seems like you’ve simply removed a crucial part of the V1 cytoarchitecture.

Thank you. R2 also pointed out this. Indeed, the model used here is a variant of the CMC model. It is the neural mass analogue of a compartmental model developed by (Jones et al., 2007). Previous work by Stephanie Jones and colleagues has validated this model using MEG data: the model has successfully explained brain dynamics including evoked responses (Jones et al., 2007), rhythmogenesis and the alpha and beta oscillations (Jones et al., 2009) and also gamma oscillations (Jones et al., 2013). Our approach is based on analysing a compartmental and a neural mass model together. To date, there is no

compartmental model based on the CMC neural mass. Thus, we could not had used the CMC model to illustrate our approach. Constructing and validating a compartmental CMC model would be a long-term project –possibly resulting in several distinct publications. Like you say, the CMC and the (Jones et al., 2007) models are very similar. The difference is in spiny stellates. This is important, however gamma oscillations that are considered here have been shown to be primarily dependent on parameters of other populations (superficial pyramidal cells and inhibitory interneurons) that are present in both models. We have added to the manuscript (p. 2):

“We combine a model of neural compartments, describing dendrites and somata introduced in [5], with a biophysical neural mass model that predicts non-invasive brain data [6]. The model of [5] had been used to explain MEG oscillations in the alpha/beta [42] and gamma bands [14]. We used statistical decision theory [7] to prove that the these two models can be combined to to infer neural dynamics in different cortical layers (laminar dynamics) using non-invasive MEG data.”

Reviewer #3:

The correlations with V1 size are interesting but come out of nowhere; something should be added to the intro about this being a feature of your analysis!

Our reply:

We apologize for this omission. We have now added on p.3:

“To validate our new approach, we asked whether we could confirm those earlier results (obtained using different computational models). We also asked whether our new model of laminar dynamics could identify the cortical layer where these differences were more pronounced. To address these questions we computed correlations between model parameters describing laminar dynamics (connections) and V1 size that is known to predict gamma peak frequency [15]. Interestingly, both datasets led to the same result.”

Reviewer #3:

Discussion:

This section does a good job of putting the results in the context of major hypotheses about gamma – however I think you should add references to the literature on the laminar generators of gamma & beta – e.g. the Traub/ Whittington /Koppel /Moghaddam type papers.

Our reply:

The reviewer is right. We have now added these references on p.8:

“Our results fit with studies that found GABAergic channels to play a prominent role in generating gamma oscillations [23] and that gamma oscillation variability reflects differences resting in GABA concentration measured with MR spectroscopy [24]. This follows a long line of work where PING and similar circuit mechanisms have been used to explain the generation of gamma oscillations [45]-[47]. Also, they confirm results we obtained earlier using different models and statistical analyses.”

Reviewer #3:

Methods & Materials:

The level of detail you describe the micro model in is much less than that of the macro/DCM model. The text suggests that the micro model is driven with a Gaussian input – in what domain – time? If so, was the micro model integrated numerically? How did you get a spectral response? Was it the same method as the DCM?

Our reply:

Sorry; we meant the Gaussian input in the time domain, indeed. We integrated the model numerically using the implicit functionality of NEURON. Then we used a simple Welch method for obtaining spectral density estimates. We added on p.11:

“ To construct the microscopic model (that we called symmetric compartmental model), we adapted NEURON code from [5]. This model can be found in <https://github.com/pinotsislab/MicroMacro/> [...] Driven with Gaussian input in the time domain [...] then simulated data from the symmetric compartmental model and made them amenable to a further DCM analysis, see *simData.mat* in the above Github repository. The model was integrated using the implicit functionality of NEURON. We then used a a simple Welch method for obtaining spectral density estimates. This was also used in DCM.”

Reviewer #3:

DCM SSR uses a parameterised noise function (*spm_csd_mtf_gu*), which includes spectral shaping (i.e. $1/f$ -like) and neuronal fluctuations in frequency space. Was the micro-model subjected to this also? If not, are the spectral outputs of each really comparable?

Our reply:

Thank you. This is an important point. Indeed, input to the two models is not identical. We now discuss this further on p.4: “This difference in superficial layers is due to the fact that the macroscale model receives explicit white and pink noise input that propagates to all layers, while the microscopic models received Gaussian spike input targeting superficial layers. Despite these

differences, we show below that SDT allows us to estimate neural mass model parameters for which the neural mass predicts the same neural dynamics as the compartmental model. This is done by fitting the neural mass to simulated data from the compartmental model.”

Reviewer #3:

A lot of effort is spent recapitulating parts of the DCM framework / equations, which emphasises how little information is provided about the microscale model methods. This balance needs addressing.

Our reply:

We apologize for this omission. We have now added more details about the microscale model etc. We have also removed some DCM equations following your suggestion below. We have now added on p.11:

“Driven with Gaussian input in the time domain, the model accurately reproduced the S1 evoked response to a tap on the hand and described the intracellular currents that give rise to signal polarity. We here used a variant of this model where we increased the number of inhibitory units from three to 10 per layer, so that their number was equal to the number of the principal cells within each mini-column. The original model comprised 10 PNs in layers 2/3, 10 PNs in layer 5, and 3 INs in both layers. The synaptic architecture followed general tenets of cortical micro-circuitry where FF connections target the granular layer and FB connections target agranular layers, see e.g. (Pinotsis et al., 2013) for a further discussion. Modelling of single neuron morphology and physiology followed (Bush and Sejnowski, 1993), using the same parameters as in (Jones et al., 2007). Details can be found in (Pinotsis et al., 2017). To ensure that relative differences in interneuron densities were accommodated, we multiplied the maximum conductance values of the corresponding connections by a factor of 0.3. To reduce the number of connection parameters (synaptic strengths) to be equal to those of the macroscopic model, we assumed that the connection weights between different mini-columns comprising the microcircuit shown in Figure 1A were the same. This assumed symmetry constraints on horizontal connectivity (within each cortical layer) of the sort assumed in mean field models that describe aggregate activity over hundreds of neurons. We then simulated data from the symmetric compartmental model and made them amenable to a further DCM analysis, see *simData.mat* in the above Github repository. The model was integrated using the implicit functionality of NEURON. We then used a simple Welch method for obtaining spectral density estimates. This was also used in DCM.”

Reviewer #3:

Equation 2 doesn't seem necessary, just cite one of the DCM papers. Or move to supplementary. Equation 4 also seems supplementary worthy. It's doesn't add much and is a distraction.

Our reply:

Sure; we have removed both equations (2) and (4) and added references as appropriate. This shortens the DCM part as also suggested by your remark above.

Reviewer #3:

Figures:

On your correlation scatter plots – both in the main text and supplementary figures – could you replace the y-label with an actual description of the parameter for readability? i.e. change ‘alpha23’ to ‘L5 DP -> L5 II’ or similar.

Our reply:

Thank you. We tried changing labels –which resulted in visually cluttered plots as there were 10 plots in each Figure. We have however added a column to Table 1 following your suggestion, that is, including an explicit description of the parameter:

Table 2 Synaptic connectivity parameters for the microscale and macroscale models.

	Description	Max conductance (μ S) Microscale model	Intrinsic connectivity Macroscale model (a.u)
a_{44}	SP->SP	0.001	4.4
a_{14}	SP->SI	0.003	4.8
a_{34}	SP->DP	0.00025	23.3
a_{41}	SI->SP	0.015	3.8
a_{31}	SI->DP	0.0003	5.9
a_{11}	SI->SI	0.0006	4.2
a_{33}	DP->DP	0.005	2.2
a_{23}	DP->DI	0.0003	4.6
a_{32}	DI->DP	0.0075	6.9
a_{22}	DI->DI	0.0006	4.16

We have also added the following on p. 4:

“These include the connection parameters between the populations shown in Figure 1: inhibitory interneurons in superficial layers, SI; pyramidal cells in superficial layers, SP; inhibitory interneurons in deep layers, DI; pyramidal cells in deep layers, DP.”

We hope that the above changes and additions to the manuscript were what you had in mind.

REVIEWERS' COMMENTS:

Reviewer #2 (Remarks to the Author):

Thank you I found the combination of the reviewers response to my comments, and indeed the other reviewers comments, greatly improved the manuscript. I have no further suggestions.

Reviewer #3 (Remarks to the Author):

I am happy that the authors have made substantial improvements to the manuscript in line with mine and the other reviewers points. The manuscript is much stronger and more detailed.